# HilE mediates motility thermoregulation in typhoidal *Salmonella* serovars at elevated physiological temperatures

**Rivka Shem-Tov**[1,2,3], **Ohad Gal-Mor**[1,2,3]*

1 The Infectious Diseases Research Laboratory, Sheba Medical Center, Tel-Hashomer, Israel, 2 Faculty of Medical & Health Sciences, Tel Aviv University, Tel Aviv, Israel, 3 Department of Clinical Microbiology and Immunology, Tel Aviv University, Tel Aviv, Israel

* Ohad.Gal-Mor@sheba.health.gov.il

## Abstract

*Salmonella enterica* is a diverse bacterial pathogen consisting of both typhoidal and nontyphoidal clinically distinct serovars. While typhoidal serovars cause in humans a systemic life-threatening enteric fever, nontyphoidal *Salmonella* (NTS) usually provoke a localized self-limiting gastroenteritis. Factors responsible for the different diseases caused by distinct *Salmonella* serovars are still poorly understood. Here, we show that at elevated physiological temperature, manifested during enteric fever (39–40°C), the transcription of the flagellar regulon, its protein translation, and flagella-mediated motility are all repressed in the typhoidal serovar, *S.* Paratyphi A. In contrast, the NTS representative serovar, *S.* Typhimurium, maintains similar or even higher levels of flagellar genes transcription, translation, and motility at 40°C relative to 37°C. By using a temperature-responsive chromogenic reporter system in conjunction with a dense transposon mutagenesis screen we found that under elevated temperature, HilE negatively regulates *S.* Paratyphi A motility in a HilD-dependent manner. Because HilD is required for the transcriptional activation of *flhDC*, encoding the master regulator of the *Salmonella* flagellar-chemotaxis regulon, null deletion of *hilE* leads to motility upregulation at elevated temperature and the loss of motility thermoregulation in *S.* Paratyphi A. Moreover, we show that a HilE-mediated motility thermoregulation is common to other typhoidal serovars, including *S.* Typhi and *S.* Sendai, but not to *S.* Paratyphi B, nor to various NTS serovars. Interestingly, the absence of HilE also leads to a hyper-uptake of *S.* Paratyphi A by THP-1 human macrophages at 40°C, but not at 37°C. Based on these results, we propose that HilE plays a unique role in motility thermoregulation in typhoidal *Salmonella* in a way that may restrain systemic dissemination of the pathogen via professional phagocytes, during the acute phase of enteric fever.

provided the original author and source are credited.

**Data availability statement:** All relevant data are within the manuscript and its Supporting Information files.

**Funding:** The work at the Gal-Mor laboratory was supported by the German-Israeli Foundation for Scientific Research and Development (GIF) under grant number I-41-416.6-2018; the Research Cooperation Lower Saxony – Israel (The Volkswagen Foundation) under grant number A128055; and the Israel Science Foundation (ISF) under grant numbers 2616/18 and 1228/23 awarded to O.G.M. The funders had no role in study design, data collection and analysis, decision to publish, or preparation of the manuscript.

**Competing interests:** The authors have declared that no competing interests exist.

## Author summary

Despite high genetic similarity, typhoidal and nontyphoidal *Salmonella* (NTS) strains of the single species *Salmonella enterica* cause in humans different diseases manifested as life-threatening enteric fever and short-term gastroenteritis, respectively. Currently, we are still ignorant about bacterial factors shaping the different lifestyles of typhoidal vs. NTS strains. Here we characterized differences in the regulation of *Salmonella* motility, which is an important virulence-associated phenotype, in response to changes in temperature, between typhoidal and NTS. We found that at elevated temperature, equivalent to the body temperature during enteric fever (39–40°C), the motility of typhoidal *Salmonella,* but not that of NTS is strongly repressed, by the negative regulator HilE in a HilD-dependent manner. Moreover, we demonstrate that HilE plays a previously unknown role in the interaction of *S.* Paratyphi A with phagocytic cells, as the absence of HilE caused enhanced uptake of this pathogen by human macrophages at elevated physiological temperature, but not at 37°C. Because motility thermoregulation by HilE was found in three different typhoidal serovars, but not in NTS, we hypothesize that motility regulation affects the interactions of *Salmonella* with its host and differences in its regulation contribute to the distinct pathogenicity of typhoidal vs. NTS strains.

## Introduction

*Salmonella enterica* (*S. enterica*) is a ubiquitous and diverse bacterial pathogen and the causative agent of different infectious diseases that range from a self-limiting gastroenteritis to a life-threatening enteric (typhoid) fever [1,2]. More than 2600 distinct *S. enterica* serovars defined so far [3] could be phenotypically classified according to their ability to infect different hosts (host-specificity profile) and the nature of the disease they cause [4]. Nontyphoidal *Salmonella* (NTS) serovars such as *S. enterica* serovar Typhimurium (*S.* Typhimurium) are generally characterized by a broad host-specificity, and their infection of immunocompetent humans normally leads to an acute inflammation of the small intestine, presented as a self-limiting gastroenteritis [5]. In contrast, Typhoidal *Salmonella* serovars, including *S. enterica* serovar Typhi (*S.* Typhi), *S. enterica* serovar Paratyphi A (*S.* Paratyphi A), and *S. enterica* serovar Sendai (*S.* Sendai) are all human-specific *Salmonella* serovars that can cause bacteremia and invasive enteric fever. In this non-inflammatory febrile disease, *Salmonella* infection does not remain contained within the intestinal mucosa, but instead is disseminated via the blood circulatory system to systemic sites, including the mesenteric lymph nodes, spleen, liver, and occasionally the gallbladder [6,7].

Besides the different host-specificity profile and the distinct clinical manifestation, another phenotype that differentiates typhoidal from NTS is motility thermoregulation. Flagella-mediated motility is an important trait that is engaged in multiple virulence-associated processes, including host cell adherence, biofilm formation, protein

secretion, and invasion [8]. Previously, we showed that while NTS retain their flagella-mediated motility at elevated physiological temperatures (39–42°C), typhoidal *Salmonella* demonstrates motility suppression under these conditions [9,10]. Nonetheless, the molecular mechanism underlying this phenotype remained elusive.

Flagellar assembly and chemotaxis in *Salmonella* is a hierarchical and tightly controlled process that involves the expression of more than 60 genes, organized in a feedforward regulatory cascade consisting of three transcriptional classes [11]. The class 1 operon contains the *flhD* and *flhC* genes that together encode the master regulator of the flagella-chemotaxis regulon, FlhDC. The heteromultimeric complex ($FlhD_4C_2$) is positioned at the top of this hierarchy and activates the transcription of class 2 operons. Class 2 genes encode components of the basal body and hook, the alternative sigma factor $\sigma^{28}$, FliA and the corresponding anti-$\sigma^{28}$ factor, FlgM. FliA activates the transcription of the class 3 genes, which encode the flagellin (FliC and FljB), other proteins needed late in flagellar biogenesis and the chemotaxis signal transduction system that directs the bacterial motion [8,12] (S1A Fig).

Here we show that motility thermoregulation in *S.* Paratyphi A and other typhoidal serovars is mediated by the negative regulator HilE, in a HilD-dependent manner. As a result, HilD-dependent *flhDC* transcription is inhibited at elevated physiological temperatures. Deletion of HilE in *S.* Paratyphi A relieves motility suppression and results in induced expression of the entire motility chemotaxis regulon and increased motility at higher temperatures. Moreover, we show that the absence of HilE significantly increases the ability of *S.* Paratyphi A to enter human phagocytic cells at elevated temperatures, but not at 37°C. These results suggest that HilE may play a role as a virulence mediator by limiting *S.* Paratyphi A uptake by phagocytic cells during the acute phase of enteric fever. This activity may maintain a balance between systemic dissemination and intestinal residing and shedding by typhoidal *Salmonella*.

## Results

### Quantifying the motility thermoregulation at the transcriptional and translational levels

Previously we reported that while *S.* Typhimurium maintained similar or even increased levels of motility at 42°C compared to 37°C, *S.* Paratyphi A demonstrated significantly reduced motility at the elevated temperature. We also showed that the reduced motility is associated with downregulation of several class 2 and 3 motility genes, including *fliZ*, *flgM*, *cheA*, and *fliC*, at 42°C relative to their expression at 37°C [9]. Here we sought to characterize this phenotype more systematically and determined the fold change in gene expression of 21 genes, representing all of the 18 functional operons, constituting the motility-chemotaxis regulon of *S.* Paratyphi A (S1A Fig). To study changes in gene expression under conditions that mimic the acute phase of enteric fever, we compared their expression at 37°C and 40°C, to reflect more accurately the most marked clinical symptom of patients with enteric fever [13,14]. We extracted RNA from *S.* Typhimurium and *S.* Paratyphi A cultures grown in LB at 37°C and 40°C and used quantitative reverse transcription polymerase chain reaction (RT-qPCR) to quantify the expression of their motility genes. In agreement with our previous findings from experiments conducted at a slightly higher temperature [9], we found that the flagellar genes were significantly downregulated in *S.* Paratyphi A at 40°C relative to their expression at 37°C (mean fold difference for 21 genes was 4.9; median fold difference was 12.5). In contrast, the expression of the motility-chemotaxis genes in *S.* Typhimurium was either similar or even moderately higher at 40°C compared to 37°C (Fig 1A).

To characterize these differences on the protein level, we applied liquid chromatography with tandem mass spectrometry (LC-MS-MS) and determined the relative levels of the flagellar proteins (S1B Fig) at 40°C relative to 37°C in *S.* Typhimurium and *S.* Paratyphi A. Concurring with the transcriptional data, we found that in *S.* Paratyphi A, flagellar and motility proteins were expressed at significantly lower levels at 40°C compared to 37°C (mean fold difference for 21 genes was 2.9; median fold difference was 13.8). The small ribosomal subunit protein uS11 RpsK that was used as an abundant temperature-independent protein control demonstrated similar expression levels at both temperatures. In contrast, the expression levels of the small heat shock protein IbpB that was used as a positive control increased by 3-fold in *S.* Typhimurium and 2-fold in *S.* Paratyphi A at 40°C relative to 37°C (Fig 1B).

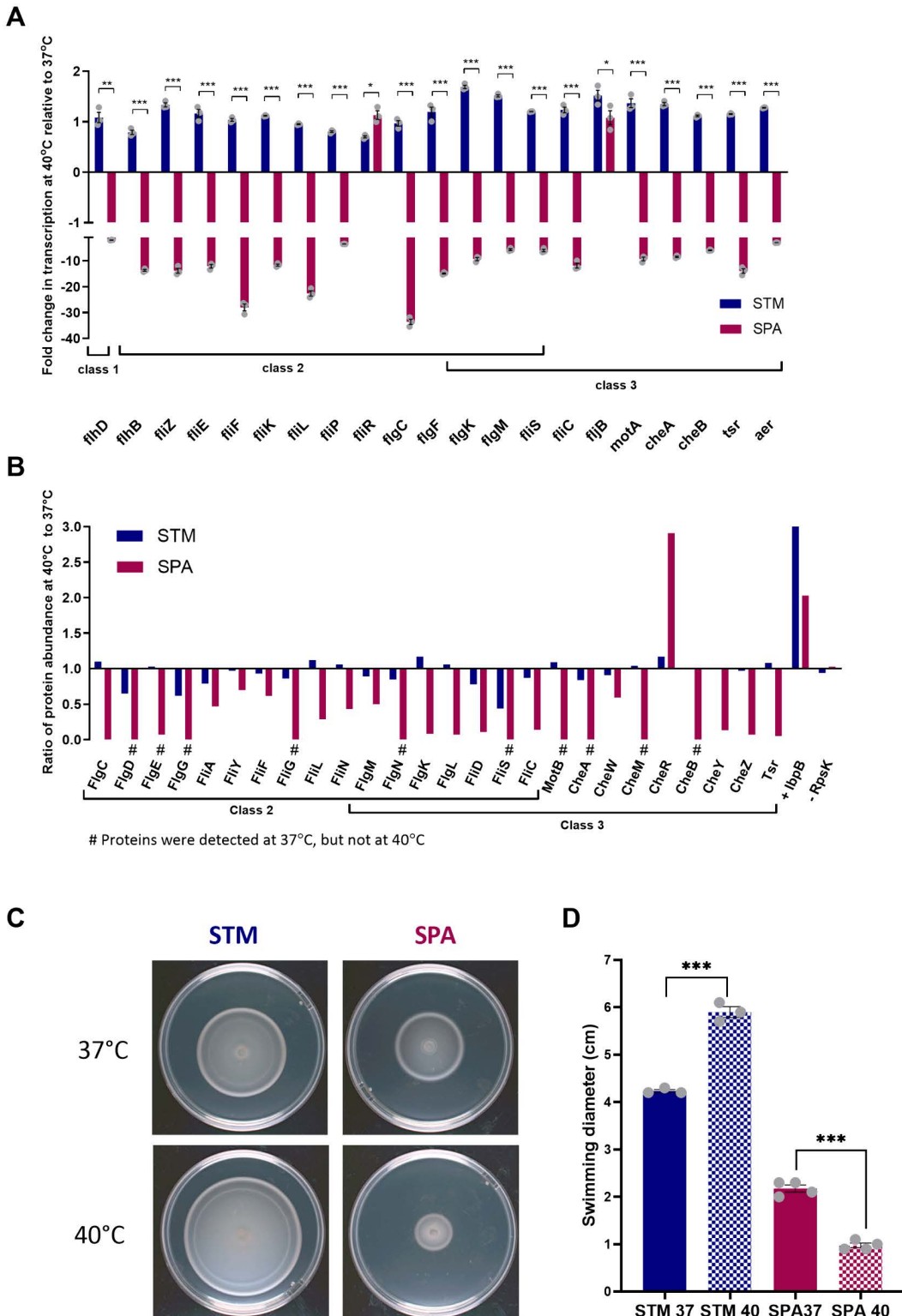

**Fig 1. Motility thermoregulation in *S.* Paratyphi A vs. *S.* Typhimurium. (A)** *S.* Typhimurium (STM) and *S.* Paratyphi A (SPA) cultures were grown in LB for 3 h at 37°C and 40°C to the late logarithmic phase under aerobic condition. The fold change in the transcription of flagellar and motility genes at 40°C relative to 37°C was determined by RT-qPCR and is normalized to the expression of the housekeeping gene *16S rRNA*. Columns show the mean

and the standard error of the mean (SEM), indicated by the error bars. Unpaired, 2-tailed Student *t* test was used to determine statistical significance between gene expression in *S*. Paratyphi A and *S*. Typhimurium. **(B)** Flagellar and motility protein expression, was determined by LC-MS/MS. Fold change in protein abundance at 40°C vs. 37°C normalized to RpoD is presented. The heat shock protein, IbpB was included as a positive control for a temperature-responsive protein. RpsK, a 30S ribosomal protein, served as a control for temperature-independent expressed protein. The # sign marks *S*. Paratyphi A proteins that were detected by MS at 37°C, but were undetected at 40°C and got a quotient value of zero. **(C)** Overnight *S*. Typhimurium and *S*. Paratyphi A cultures that were grown at 37°C were spotted on soft LB agar plates that were incubated at 37°C or 40°C for 5.5 h. **(D)** The measured swimming radius of *S*. Typhimurium and *S*. Paratyphi A cultures at 37°C and 40°C is shown. Bars indicate the mean value and its associated SEM of at least three biological repeats. Unpaired, 2-tailed Student *t* test was used to calculate statistical significance between the swimming radius at 37°C and 40°C.*, P-value <0.05; **, P-value <0.01 ***, P-value < 0.0001.

As expected, these differences were also reflected at the phenotypic level, as the swimming motility of *S*. Paratyphi A on LB soft agar plates was reduced by more than 2-fold at 40°C compared to 37°C, while *S*. Typhimurium motility slightly increased by 1.4-fold under these conditions (Fig 1C and 1D). We concluded from these results that the motility regulon in *S*. Paratyphi A, but not in *S*. Typhimurium, is repressed by an unknown mechanism at 40°C compared to 37°C, and that this repression is evident on the transcriptional, translational, and phenotypic levels.

### Construction of a temperature-responsive reporter system of the *S*. Paratyphi A motility regulon

To identify genes involved in the motility thermoregulation in *S*. Paratyphi A we sought to take a transposon mutagenesis approach and designed a temperature-responsive reporter system that allowed visual screening of a large transposon library. For that purpose, a promoterless β-galactosidase gene (*lacZY*) was cloned under three different motility gene promoters, which demonstrated in *S*. Paratyphi A a high fold difference in their expression between 37°C and 40°C, including *motA* (9-fold), *fliC* (12-fold), and *flhB* (17-fold). These constructs were introduced into *S*. Paratyphi A and the appearance of these reporter strains was visually evaluated on LB agar plates supplemented with 5-bromo-4-chloro-3-indolyl-D-galactoside (X-gal) at 37°C and 40°C. As the color of the colonies is dependent on the activity of the *lacZ* gene product, we reasoned that significant changes in the activity of the cloned promoters would result in a chromogenic change between 37°C and 40°C. This analysis in *S*. Paratyphi A indicated that while the *fliC::lacZ* reporter demonstrated dark blue color and the *motA::lacZ* reporter showed very bright color at both temperatures, the *flhB::lacZ* reporter exhibited clear interchanging temperature-dependent colony color, with a blue appearance at 37°C, but a colorless appearance at 40°C (Fig 2A). In contrast to *S*. Paratyphi A, *flhB::lacZ* expression in *S*. Typhimurium demonstrated similar blue color at both temperatures, suggesting comparable levels of promoter activity in this background (Fig 2B). Indeed, a quantitative β-galactosidase assay using *S*. Typhimurium and *S*. Paratyphi A cultures expressing the *flhB::lacZ* reporter demonstrated constant and high β-galactosidase activity in *S*. Typhimurium at both temperatures, but significantly lower expression in *S*. Paratyphi A at 40°C compared to its high activity at 37°C (Fig 2C), in agreement with the above RT-qPCR and proteomics data. Therefore, we chose to continue the subsequent screening with the *flhB::lacZ* reporter, which demonstrated temperature-dependent promoter activity that was reliably correlated with color changes in its colony appearance on LB-X-gal plates.

### Genetic screen to identify motility thermoregulators in *S*. Paratyphi A

Based on these results, we designed a genetic screen of a Tn-mutant library of a *S*. Paratyphi A strain harboring the *flhB::lacZ* reporter system. As the colonies of this reporter strain appeared transparent at 40°C, due to the repression of the *flhB* promoter, we expected to identify blue colonies in insertion mutants that would lose their repressing thermoregulation. A dense transposon library was generated by conjugating an *Escherichia coli* SM10 λ *pir* donor strain carrying the pJA1 plasmid into a nalidixic acid-resistant *S*. Paratyphi A strain, carrying the *flhB::lacZ* reporter system. pJA1 harbors an inducible mini-Tn10 transposon that has a decreased integration bias [15]. Transconjugant *S*. Paratyphi A cells were IPTG-induced and plated on X-gal LB agar plates under the selection of ampicillin (to select for the reporter plasmid), kanamycin (to select for the transposon), and nalidixic acid (to select for *S*. Paratyphi A) at 40°C. Overall, we screened 56,200 *S*. Paratyphi A

**A**

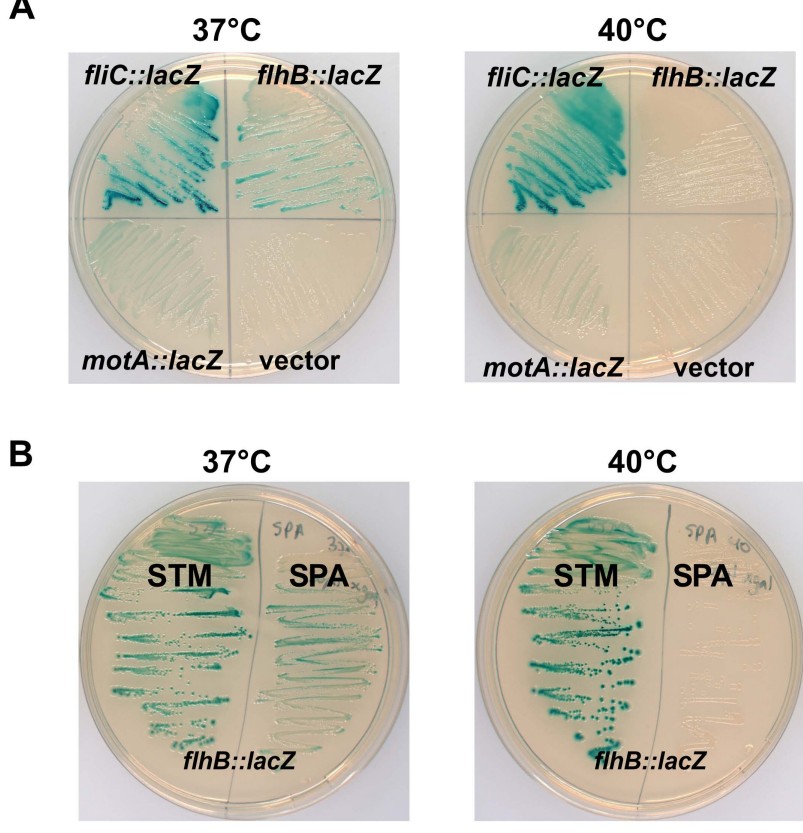

**B**

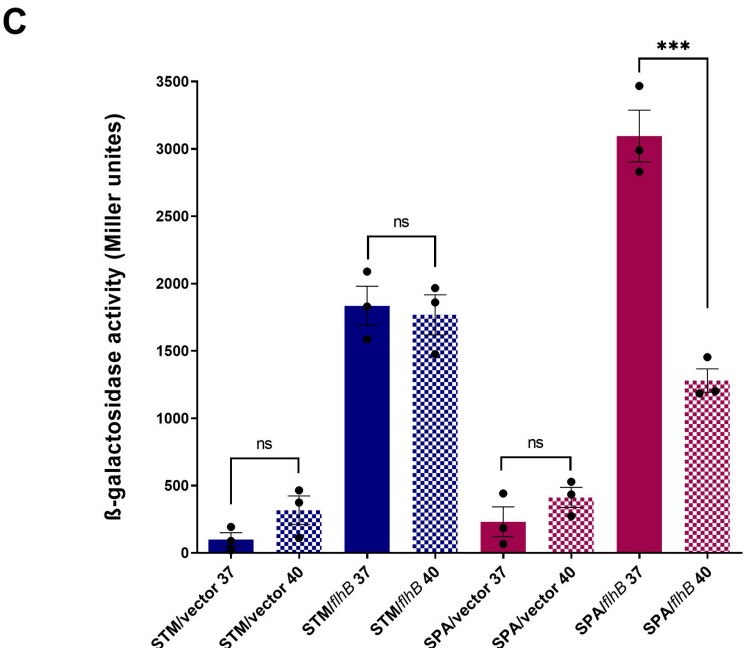

**C**

**Fig 2. Construction of a temperature-responsive reporter system for the *S*. Paratyphi A motility regulon. (A)** The *fliC*, *motA* and *flhB* gene promotors from *S*. Paratyphi A 45157 were cloned upstream to the β-galactosidase gene (*lacZY*) in a low-copy number vector. *S*. Paratyphi A strains expressing the empty vector only (pMC1403) or each one of the three reporters were steaked onto LB/ X-gal agar plates that were incubated at 37°C

and 40°C. Blue colonies indicate high expression of the promoters, while transparent colonies indicate low/no expression. **(B)** *S*. Typhimurium (STM) and *S*. Paratyphi A (SPA) strains expressing the reporter system *flh*B::*lacZ* were streaked onto LB/X-gal and incubated at 37°C and 40°C. Differences in the expression of this construct can be appreciated by the blue/ transparent color of the colonies developed under each condition **(C)** β-galactosidase specific activity derived from the expression of *flhB*::*lacZ* was quantified in *S*. Typhimurium and *S*. Paratyphi A cultures grown at 37°C and 40°C. The empty vector (pMC1403) was included as a negative control that shows the basal (leaky) *lacZ* expression. The mean of three biological repeatsis shown, while the error bars represent the SEM. Unpaired, 2-tailed Student *t* test was used to determine the statistical significance between β-galactosidase activity at 37°C and 40°C.***, P-value < 0.001; ns, not statistically significant.

Tn-mutant colonies in three independent screening rounds, which constitutes about 11-fold coverage of the *S*. Paratyphi A genome. Overall, twenty-three colonies with a blue color appearance at 40°C were selected for further characterization (Fig 3A). These clones were streaked on LB agar X-gal plates and incubated overnight at 37°C and 40°C for visual evaluation of their *flhB::lacZ* expression. Fifteen out of the twenty-three examined colonies maintained their blue color at both temperatures (Fig 3B), indicating a stable *flhB::lacZ* expression at elevated temperature and a possible loss of their motility thermoregulation. To further quantify the activity of the *flhB::lacZ* reporter in these mutants, we applied a quantitative β-galactosidase assay for mutants grown at 40°C and calculated the β-galactosidase activity in these mutants relative to its expression in the WT background. As shown in Fig 3C, in ten out of the fifteen Tn-mutants tested, we found an elevated β-galactosidase activity at 40°C compared to the WT, indicating an induced *flhB* promoter activity under conditions, in which it is normally repressed in the WT background. To confirm the possible loss of motility thermoregulation, we further analyzed their swimming motility on soft agar LB plates at 40°C vs. 37°C (Fig 3D-3E). Strikingly, we found that in eight out of the fifteen mutants examined (mutants 1–2, 3–2, 4–2, 8–2, 3–3, 9–3, 15–3, and 19–3), the *flhB::lacZ* expression and motility were significantly induced compared to the WT background at 40°C, consistent with a possible loss of their motility thermoregulation.

To identify the transposon insertion sites, we applied two-step PCR using mini-Tn10 transposon-specific and random primers (see Materials and Methods), followed by Sanger sequencing of the resulting amplicons. The identified Tn-mutated defective genes are listed in Table 1 and include genes from the *Salmonella* pathogenicity island (SPI)-1 (*hilE, invG,* and *sipD*) and lysine- diaminopimelic acid (DAP) biosynthesis (*dapD* and *dapB*) regulons.

To confirm the role of these genes in *S*. Paratyphi A motility thermoregulation, non-polar, in-frame deletions of *dapB, invG,* and *hilE* were constructed in a "clean" *S*. Paratyphi A 45157 background. While null mutations in *dapB, invA,* and *invG* were not confirmed to be involved in *S*. Paratyphi A motility thermoregulation, *hilE* deletion, which was hit twice at two different insertion sites, and at two orientations was shown to relieve motility suppression at 40°C to a similar extent as found in the Tn-mutants 3–2 and 8–2 (S2 Fig). We therefore concluded from these experiments that in *S*. Paratyphi A, HilE may play a role as a negative thermoregulator that represses *S*. Paratyphi A motility at elevated physiological temperatures.

## HilE thermoregulates motility in *S*. Paratyphi A

To better understand the mechanism by which HilE affects thermoregulation in *S*. Paratyphi A we studied the transcription of twenty-one motility-chemotaxis genes in *S*. Paratyphi A WT and Δ*hilE* null mutant strains grown at 40°C. We found that in the absence of HilE, most of the studied genes exhibited significantly higher transcription relative to the WT. Complementing Δ*hilE* from a low-copy number vector (pWSK29::*hilE*), largely downregulated the expression of the motility genes to similar levels as in the WT background (Fig 4A). In agreement, Western blot (WB) analysis using an α- *S*. Paratyphi A FliC antibody demonstrated higher levels of FliC in the absence of *hilE* at 37°C and 40°C compared to the WT background. Complementing Δ*hilE* expression ectopically resulted in decreased FliC expression to levels similar to or even lower than those found in the WT (Fig 4B). Finally, we examined the swimming motility phenotype at 37°C and 40°C. As shown in Fig 4C and 4D, while *hilE* deletion significantly increased *S*. Paratyphi A motility at 40°C, HilE complementation restored the swimming motility to similar levels as in the WT strain. These results indicated that HilE functions as a key thermoregulator in *S*. Paratyphi A by suppressing motility gene expression at elevated physiological temperatures.

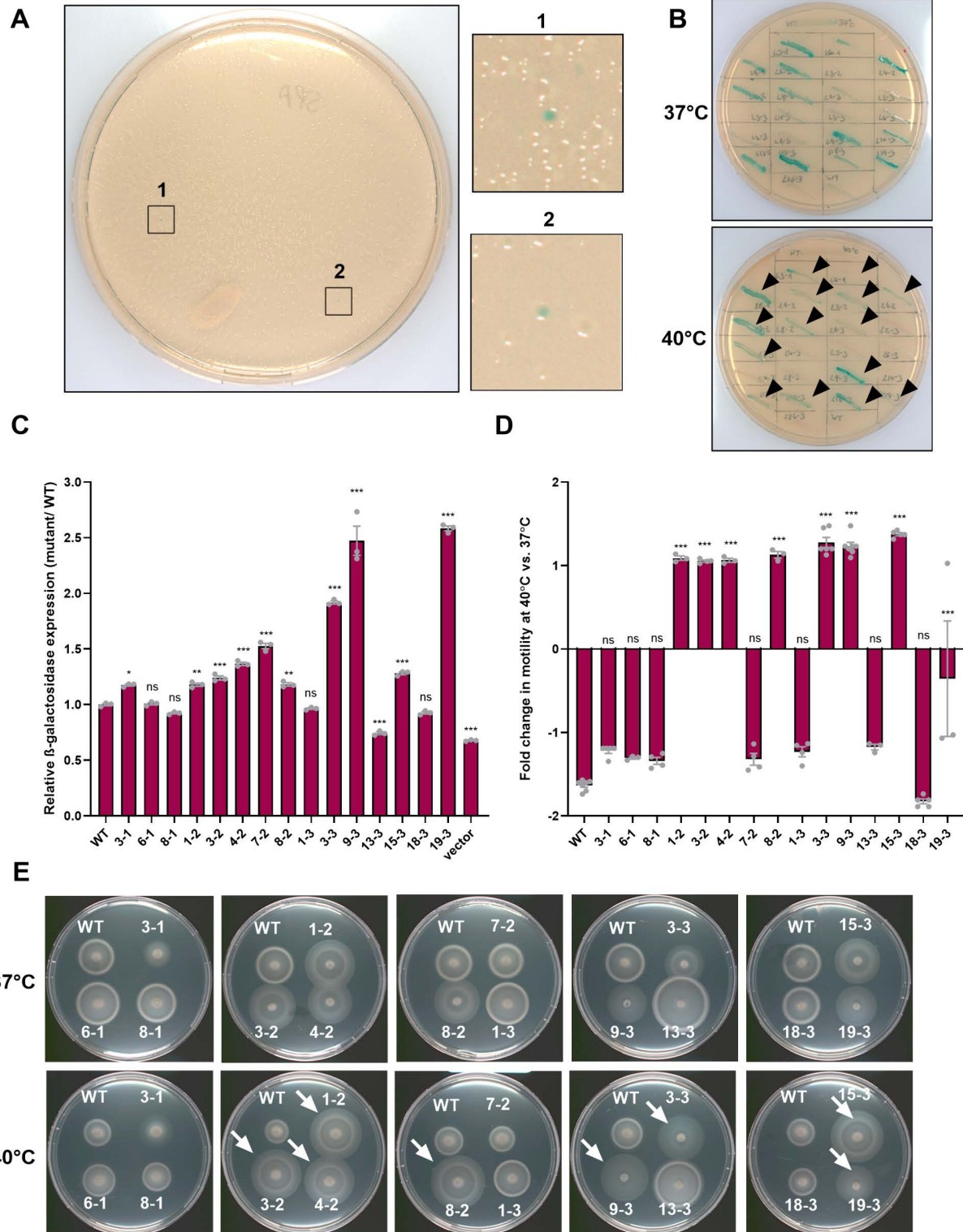

**Fig 3. Genetic screen to identify motility thermoregulators in *S*. Paratyphi A. (A)** A dense mini-Tn10 transposon library in a *S*. Paratyphi A strain expressing *flhB::lacZ* was plated on LB/X-gal agar plates that were incubated at 40°C. Tn-mutants with induced *flhB::lacZ* expression were identified based on their blue appearance. **(B)** Twenty-three Tn-mutants that presented blue color at 40°C were passed onto LB/X-gal plates and incubated at

37°C and 40°C. Fifteen mutants that maintained high expression of *flhB::lacZ* at 40°C are indicated by black arrowheads. **(C)** The relative change in the β-galactosidase activity of the *flhB::lacZ* between the WT and fifteen Tn-mutants was determined at 40°C and is presented relative to the expression of *flhB::lacZ* in the WT background. Basal *lacZ* expression of *S*. Paratyphi A harboring pMC1403 (vector) was used as a negative control. One-way ANOVA with Dunnett's multiple comparisons test was used to determine statistical significance in relation to *S*. Paratyphi A WT. *, P-value <0.05; **, P-value <0.01; ***, P-value < 0.001; ns, not statistically significant. **(D)** Fold change in swimming motility at 40°C relative to their motility at 37°C was calculated for the parental WT strain and its isogenic Tn-mutants. One-way ANOVA was used to determine statistical significance relative to the *S*. Paratyphi A WT background. **(E)** Motility swimming of *S*. Paratyphi A WT and its fifteen isogenic Tn-mutants was determined on soft agar plates that were incubated at 37°C and 40°C for 5.5 h. Mutants that demonstrated higher motility at 40°C relative to the WT background are indicated by white arrows.

**Table 1. Transposon mutants and insertion sites.** The Tn insertion site of eight S. Paratyphi A mutants that presented loss of motility thermoregulation (Fig 3) was determined by Sanger sequencing. Mutant number, screen number, the affected genes, the insertion position of the Tn and their annotation are shown.

| Mutant no. | Screen no.[1] | Affected gene | Insertion position[2] | Tn. direction | Accession no. | Description | Note |
|---|---|---|---|---|---|---|---|
| 1 | 2 | *sipD* | +970 | 3' < -----5' | XBC73250.1 | SPI-1 type III secretion system needle tip complex protein SipD | Part of operon *sipABCDA* |
| 3 | 2 | *hilE* | +259 | 5'-----> 3' | XBC75704.1 | A negative regulator of HilD | |
| 4 | 2 | *yffB* | +63 | 5'-----> 3' | XBC75287.1 | ArsC family reductase | Part of operon *yffB-dapD* |
| 8 | 2 | *hilE* | +347 | 3' < -----5' | XBC75704.1 | A negative regulator of HilD | |
| 3 | 3 | *invG* | +541 | 5'-----> 3' | XBC73235.1 | type III secretion system outer membrane ring protein InvG | Part of operon *invHFGEABC* |
| 9 | 3 | *dapB* | -56 | 5'-----> 3' | XBC75575.1 | 4-hydroxy-tetrahydrodipicolinate reductase | The entry site at the gene promotor |
| 15 | 3 | *dgoT* | +80 | 5'-----> 3' | XBC76299.1 | MFS transporter | |
| 19 | 3 | *dapB* | -56 | 5'-----> 3' | XBC75575.1 | 4-hydroxy-tetrahydrodipicolinate reductase | The entry site at gene promotor |

[1]No relevant mutants were found in screen number 1.

[2]Insertion position of the transposon from the first methionine codon of the hit ORF

### HilE thermoregulates *flhDC* expression in a HilD-dependent manner

HilE was previously shown to negatively regulate HilD, the main transcriptional activator of SPI-1, by binding to HilD and preventing HilD homodimerization [16]. HilD homodimer is known to bind directly upstream to *flhDC* at promoter five (P5), in a way that activates *flhDC* transcription [17]. Based on this information, we hypothesized that in *S*. Paratyphi A, motility thermoregulation is mediated by HilE via HilD, in a way that prevents HilD from binding to the *flhDC* promoter and thereby inhibits the expression of the entire motility regulon. To test this hypothesis, we constructed a transcriptional fusion between the *flhDC* promoter and *lacZ* and studied how the absence of HilE and HilD affects *flhDC* transcription at 37°C and elevated physiological temperatures. As shown in Fig 5A, the expression of *flhD::lacZ* was lower in *S*. Paratyphi A when grown at 40°C compared to 37°C, by about 3-fold. Knocking out *hilD* resulted in a significant decrease in *flhD::lacZ* expression at 37°C, indicating that HilD is required for the transcription of the motility-chemotaxis master activator, FlhDC. Interestingly, a null Δ*hilE* deletion significantly increased *flhD::lacZ* expression at 37°C as well as at 40°C. However, a double deletion of Δ*hilE hilD* counteracted this upregulation and resulted in expression levels similar to the expression observed in the Δ*hilD* mutant strains (Fig 5A).

Similarly, using RT-qPCR, we determined the transcriptional levels of *flhD*, *flhB* and *fliC* as representative genes from classes 1, 2, and 3 of the flagellar regulon, respectively. In agreement with the β-galactosidase results, RT-qPCR demonstrated that while the lack of *hilE* led to a significant increase in the expression of *flhD*, *flhB* and *fliC* at 40°C, the absence

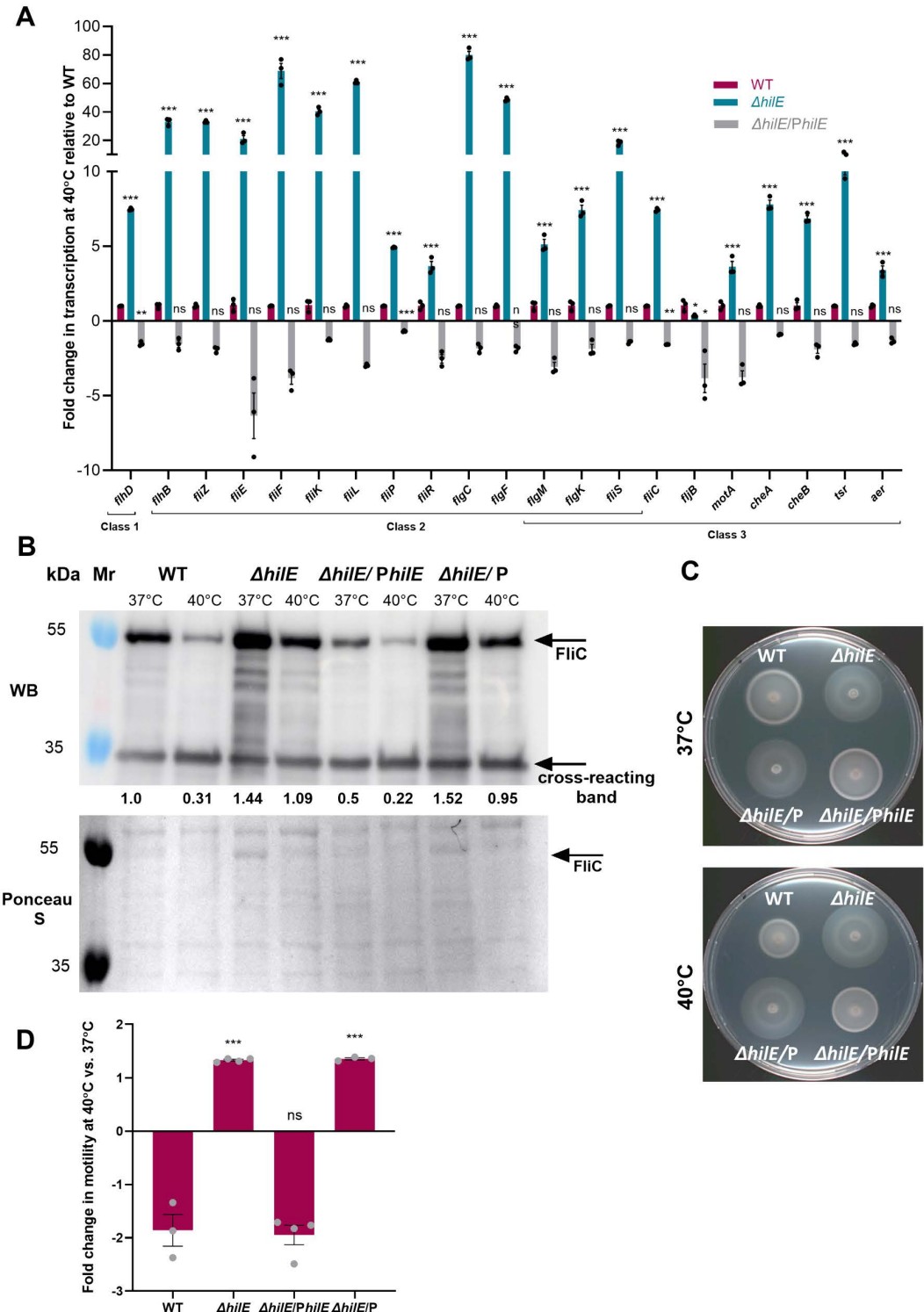

**Fig 4. HilE thermoregulates motility in *S*. Paratyphi A. (A)** Total RNA was extracted from *S*. Paratyphi A WT, Δ*hilE*, and Δ*hilE* expressing *hilE* from a low copy-number plasmid as a complemented strain (Δ*hilE*/P*hilE*). RT-qPCR was used to determine changes in the expression of 21 flagellar-chemotaxis genes in the mutant strains relative to the WT background. The mean change in expression of three biological repeats is shown and its associated SEM is represented by the error bars. One-way ANOVA was used to determine statistical significance relative to *S*. Paratyphi A WT. **(B)** *S*.

Paratyphi A WT, its isogenic Δ*hilE* null mutant strains and Δ*hilE* complemented with either *hilE* (P*hilE*) or the vector (pWSK29) only were grown for 3 h at 37°C and 40°C to the late logarithmic phase, under aerobic conditions. The level of FliC at the cellular fraction was analyzed by Western Blotting using an anti- *S*. Paratyphi A FliC antibody. Ponceau S staining and the detection of a ~34 kDa cross-reacting band were used as loading controls. Quantification of the FliC band intensities relative to its intensity in the WT background at 37°C was calculated by ImageJ and is shown under the WB panel. **(C)** Swimming motility of the indicated *S*. Paratyphi A strains was evaluated on soft agar plates that were incubated at 37°C and 40°C for 6 h. **(D)** The fold change in the motility radius at 40°C vs. 37°C in at least three assays is shown. One-way ANOVA was used to determine statistical significance relative to *S*. Paratyphi A WT. ***, P-value < 0.001; ns, not statistically significant.

of both *hilE* and *hilD* negated this effect and caused a decrease in gene expression, to similar or even lower levels than the WT background (Fig 5B).

Western blotting, using α- *S*. Paratyphi A FliC antibody, further confirmed these results on the protein level and showed lower levels of FliC in the WT strain at 40°C compared to 37°C (about 20%). In the Δ*hilE* background the repressed expression of FliC at 40°C was largely relieved and the already high expression of FliC at 37°C was even further induced. In the absence of HilD, FliC expression decreased at both temperatures and the increase in FliC expression at 40°C in the absence of HilE was lost in the Δ*hilE hilD* background.

Finally, we studied the swimming motility phenotype at 37°C and 40°C. As shown in Fig 5D, while Δ*hilE* significantly elevated *S*. Paratyphi A motility, the Δ*hilE hilD* double mutant completely neutralized the effect of *hilE* deletion and displayed motility levels similar to those of the WT.

Accumulatively, we concluded from these experiments that the motility thermoregulation in *S*. Paratyphi A is mediated by the repression of the motility-chemotaxis genes by HilE at elevated temperatures in a HilD-dependent manner, possibly via the interchanging binding of HilD to the *flhDC* promoter, which is regulated by the protein-protein interactions with HilE, as was shown previously [16].

## HilE thermoregulates motility in other typhoidal *Salmonella*

Next, we asked whether motility thermoregulation characterizes other serovars of *S. enterica*. To that end, we studied the motility of five highly prevalent NTS (Typhimurium, Enteritidis, Infantis, Muenchen, and Newport) and four typhoidal (Typhi, Paratyphi A, Paratyphi B, and Sendai) at 37°C compared to 40°C. Remarkably, we found that while the motility of all studied NTS serovars increased at 40°C relative to 37°C, in the typhoidal serovars Paratyphi A, Typhi, and Sendai, but not Paratyphi B, swimming motility was reduced at 40°C relative to 37°C (Fig 6A). Repeating this experiment with a wider array of 16 NTS serovars and at 42°C (under the temperature our original observation was made [9]), further confirmed these results and demonstrated similar motility of all NTS serovars at 42°C and 37°C, but significantly impaired motility of typhoidal strains at the elevated temperature relative to 37°C (S3 Fig).

We concluded from these results that motility thermoregulation is common to three typhoidal *Salmonella* serovars, but not to *S*. Paratyphi B, and that motility is not thermoregulated in the same way in NTS.

To investigate if the role that HilE plays in motility thermoregulation is unique to *S*. Paratyphi A or is common to other typhoidal serovars, we constructed a null non-polar *hilE* deletion in *S*. Typhi and *S*. Sendai. Comparing the swimming motility of the WT backgrounds to their isogenic Δ*hilE* mutants showed that the absence of HilE in serovars Paratyphi A, Typhi, and Sendai relieved motility suppression at 40°C and led to significantly increased swimming levels than the WT background at 40°C (Fig 6B and 6C). We concluded from these experiments that HilE plays a similar role as a motility thermoregulator in all three typhoidal *Salmonella* serovars, but not in *S*. Typhimurium, as an NTS representative serovar.

## HilE affects *S*. Paratyphi A uptake by human macrophages at elevated temperatures

To place the above findings in a physiological context, we next studied the role of HilE in macrophage uptake. After crossing the epithelial gut barrier, *Salmonella* are taken up by phagocytic cells, including macrophages that disseminate

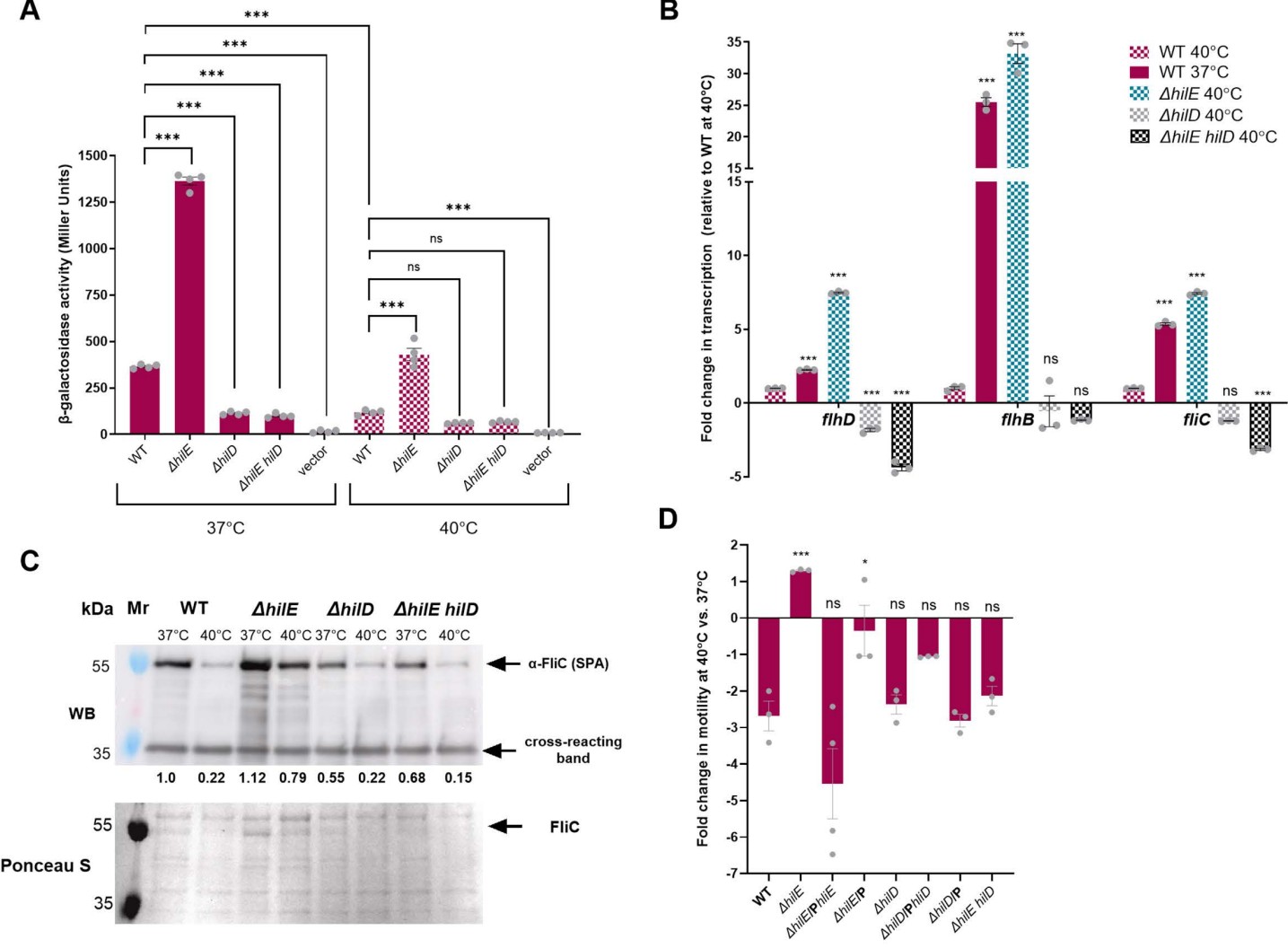

**Fig 5. HilE thermoregulates *flhDC* expression in a HilD-dependent manner. (A)** β-galactosidase activity (shown in Miller Units) of *S*. Paratyphi A WT strain and its isogenic null mutant strains Δ*hilE,* Δ*hilD* and Δ*hilE hilD* harboring the *flhD*::*lacZ* reporting construct. Bacterial cultures were grown at 37°C and 40°C and assayed for β-galactosidase activity as explained in the Material and Methods section. Basal *lacZ* expression in *S*. Paratyphi A harboring the cloning vector pMC1403 (vector) was used as a negative control. One-way ANOVA with Dunnett's multiple comparisons test was used to determine statistical significance relative to the WT at each temperature, and Unpaired, 2-tailed Student *t* test was used to determine statistical significance between the WT strains grown at 37°C and 40°C. **(B)** Fold change in the transcription level of the flagellar genes *flhD* (class 1), *flhB* (class 2) and *fliC* (class 3) was determined in the *S*. Paratyphi A WT and its isogenic Δ*hilE*, Δ*hilD* and Δ*hilE hilD* mutant strains and normalized to the housekeeping gene *16S rRNA*. Total RNA was extracted from cultures that were grown for 3 h at 37°C and 40°C to their late logarithmic phase, under aerobic conditions. The mean change in expression of three repeats is shown, with SEM represented by the error bars. One-way ANOVA was used to determine statistical significance in relation to *S*. Paratyphi A WT at 40°C. **(C)** The level of FliC at the cellular fraction was analyzed by Western Blotting, using an anti-SPA FliC antibody. Proteins were extracted from *S*. Paratyphi A WT and its isogenic Δ*hilE*, Δ*hilD*, and Δ*hilE hilD* null mutant strains that were grown at 37°C and 40°C as above. Ponceau S staining and the detection of a ~ 34kDa cross-reacting band were used as loading controls. The intensities of the FliC bands relative to its intensity in the WT background at 37°C is shown under the WB panel. **(D)** *S*. Paratyphi A WT, its isogenic null mutants (Δ*hilE*, Δ*hilD* and Δ*hilE hilD*) and mutants strains complementing with *hilE* (P*hilE*), *hilD* (P*hilD*) or the cloning vector (pWSK29) only (P) were grown overnight at 37°C and then spotted on a soft agar plates that were incubated at 37°C and 40°C for 7 h. The fold change in their motility at 40°C vs. 37°C is shown, with the SEM represented by the error bars. One-way ANOVA was used to determine statistical significance relative to the motility of *S*. Paratyphi A WT. *, P-value <0.05; ***, P-value<0.001; ns, not statistically significant.

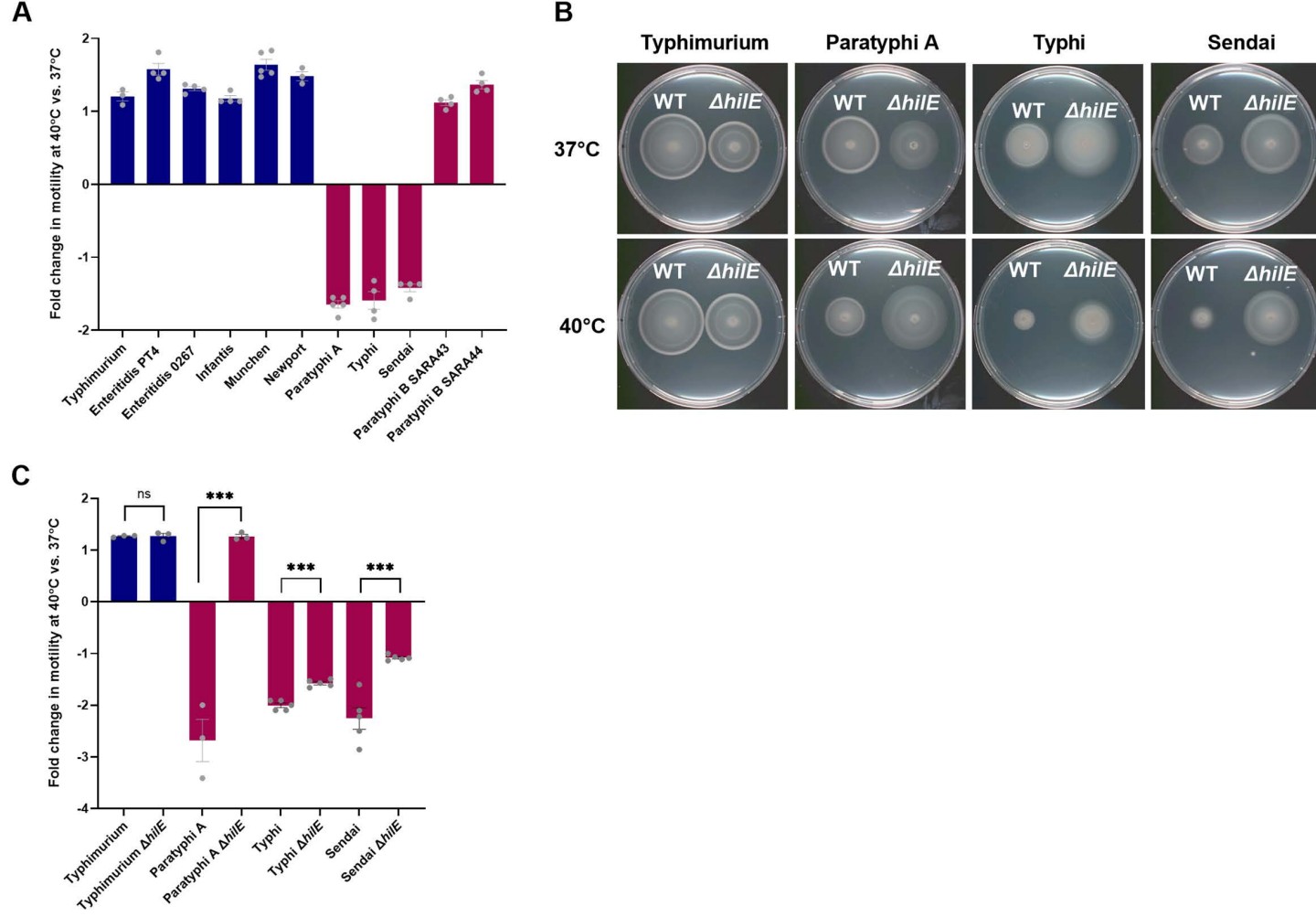

**Fig 6. HilE thermoregulates motility in other typhoidal *Salmonella*. (A)** Prevalent NTS serovars (Typhimurium, Enteritidis, Infantis, Muenchen, and Newport) and typhoidal *Salmonella* serovars (Typhi, Paratyphi A, Paratyphi B, and Sendai) cultures were grown overnight at 37°C and then spotted on soft agar plates that were incubated at 37°C and 40°C. The motility radius was measured for *S.* Paratyphi B strains after 4.5 h; *S.* Enteritidis after 5 h; *S.* Muenchen and *S.* Newport after 5.5 h; *S.* Typhimurium and *S.* Infantis after 6 h; and *S.* Paratyphi A, *S.* Typhi and *S.* Sendai after 7 h. The fold change in the motility radius at 40°C vs. 37°C is shown, while the error bars indicate the SEM. **(B)** WT and isogenic Δ*hilE* null mutant strains of *S.* Typhimurium, *S.* Paratyphi A, *S.* Typhi, and *S.* Sendai were grown overnight at 37°C and spotted onto soft agar plates that were imaged after incubation at 37°C and 40°C. **(C)** The fold change in motility swimming at 40°C vs. 37°C is shown. The motility radius was measured for *S.* Typhimurium after 5 h; *S.* Paratyphi A 7 h; *S.* Sendai 8.5 h; and *S.* Typhi 12 h. Unpaired, 2-tailed Student *t* test was used to determine statistical significance between WT and its isogenic Δ*hilE* mutants. ***, P-value < 0.001; ns, not statistically significant.

the pathogen to systemic sites [18]. Given that macrophages are major players in early immunity to *Salmonella* and their response could influence the course of the infection, we sought to study the role of HilE in *Salmonella* uptake in THP-1 human macrophages. To this end, cultures of serovars Typhimurium, Paratyphi A, Typhi and Sendai grown overnight at 37°C and 40°C were used to infect differentiated THP-1 cells that were also incubated at 37°C or 40°C, at multiplicity of infection (MOI) ~10 (bacteria per macrophage). Two h post-infection (p.i.), the cells were washed, and gentamycin was added to kill extracellular *Salmonella*. In agreement with our [10] and other [19] previous studies, which showed a correlation between motility and macrophages uptake, we found that *S.* Typhimurium that presented elevated motility at 40 vs. 37°C, also showed increased macrophage uptake at the elevated temperature. In contrast, *S.* Paratyphi A that was

motility-impaired at 40°C demonstrated decreased uptake under this condition compared to 37°C. To study the effect of HilE on *Salmonella* uptake we also compared *Salmonella* entry to macrophages by the WT and its Δ*hilE* isogenic mutant. Interestingly, we found that while the Δ*hilE* mutant in *S.* Typhimurium presented similar uptake to the WT background at both tested temperatures, the absence of HilE in *S.* Paratyphi A prominently improved its uptake by more than 10-fold at 40°C, but not at 37°C. Higher levels of macrophages uptake at elevated temperature were also identified in the HilE mutants of the other typhoidal serovars *S.* Typhi and *S.* Sendai, when we compared the uptake of their Δ*hilE* background to the WT at 40°C (Fig 7A). Nonetheless, these differences although reached statistical significance were less pronounced than in *S.* Paratyphi A, most likely due to their overall much slower motility at both temperatures (Fig 7B).

We hypothesized that the superior uptake of *S.* Paratyphi A in the Δ*hilE* mutant was mediated by its increased motility and/ or flagella overexpression at 40°C. To examine this possibility, we subsequently studied its uptake in a Δ*fliC* (encoding the structural protein of the flagella) mutant strain and at lower MOI of one. As expected, the lack of flagellin significantly reduced *S.* Paratyphi A uptake at 37°C and the Δ*hilE fliC* double mutant strain no longer demonstrated improved uptake at 40°C, as the Δ*hilE* mutant did (Fig 7C). Moreover, conducting these experiments with and without a mild centrifugation step indicated that centrifugation, although increased the overall uptake of the strains, did not change much the enhanced uptake of the Δ*hilE* mutant or the effect of the temperature (S4 Fig). These results suggested that the contribution of HilE in *S.* Paratyphi A uptake goes behind merely the improvement in bacteria motility and is likely connected to additional roles of the flagella in host cell interactions, such as flagella-mediated adherence [10,20].

Overall, we concluded from these experiments that HilE affects *Salmonella* interaction with human phagocytic cells and modulates typhoidal *Salmonella* virulence-associated phenotypes in a temperature-dependent manner. Under elevated temperature, the absence of *hilE* enhances flagella expression, motility and human macrophages uptake of *S.* Paratyphi A and possibly other typhoidal serovars.

## Discussion

Bacterial motility represents a critical virulence-associated phenotype of many bacterial pathogens, providing the capacity to navigate toward favorable environments and away from hostile conditions [21]. This locomotive ability serves multiple functions essential for host-pathogen interactions, including chemotaxis toward nutrient sources, gaining access to host epithelial cells, colonization of preferred tissues, biofilm formation, and immune evasion [22–24].

In *S. enterica*, motility plays particularly crucial roles throughout the infection cycle as flagellar-mediated swimming is essential for *Salmonella* to penetrate the mucus layer protecting epithelial surfaces, facilitates the initial approach and attachment to intestinal epithelial cells, [25], and is required for *Salmonella*'s capacity to invade M cells in Peyer's patches [26,27]. Therefore, previous studies have demonstrated that non-motile *Salmonella* mutant strains exhibit significantly attenuated virulence in both *in vitro* and *in vivo* models [19,28].

Due to the substantial metabolic investment required for flagella assembly and flagellar-mediated motility [29], *Salmonella* employs sophisticated mechanisms to modulate its motility in response to environmental cues encountered during host infection [8]. Current knowledge indicates that *Salmonella* interprets a diverse array of environmental signals to optimize flagellar expression and function. For example, oxygen availability significantly influences *Salmonella* motility, with aerobic conditions generally promoting flagellar expression through the ArcA/ArcB two-component system [30]. Other studies have demonstrated that changes in the intracellular pH [31] and the presence of short-chain fatty acids produced by the gut microbiota [32] also modulate *Salmonella* motility.

While most of these regulation studies have relied on *S.* Typhimurium as a model for the entire *S. enterica* species, previous reports from our group have shown differences in the way host cell invasion and motility are regulated between *S.* Typhimurium and *S.* Paratyphi A [9,33,34]. These observations have also led us to speculate that a distinct regulatory setup of motility and invasion may be associated with the different lifestyles and disease manifestations caused by NTS vs. typhoidal serovars. One of the most striking differences we found was that under elevated physiological temperature

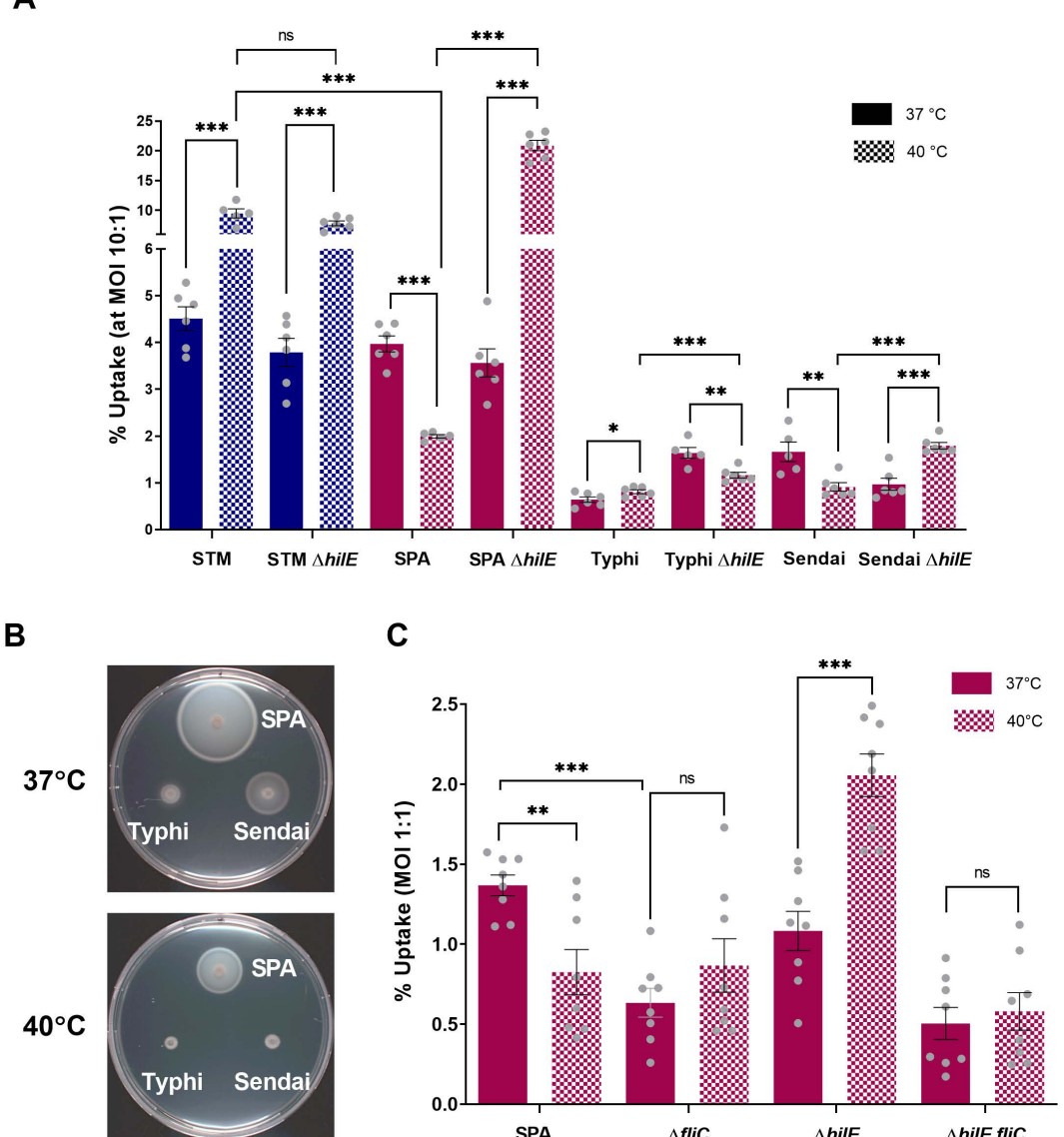

**Fig 7. HilE controls *S*. Paratyphi A uptake by human macrophages at elevated temperature. (A)** Cultures of *S*. Typhimurium, *S*. Paratyphi A, *S*. Typhi and *S*. Sendai and there isogenic *hilE* null mutant strains were grown at 37 or 40°C and were used to infect differentiated THP-1 human macrophage-like cells at MOI of ten at both temperatures. *Salmonella* uptake by THP-1 cells was determined using the gentamicin protection assay and calculated the percentage of the intracellular CFUs recovered at 2 h p.i from the total number of CFUs used to infect the cells. Bars present the mean value of six biological repeats and its calculated SEM. Unpaired, 2-tailed Student *t* test was used to calculate statistical significance. **(B)** Overnight cultures of *S*. Paratyphi A, *S*. Typhi and *S*. Sendai grown at 37°C were spotted on soft LB agar plates that were incubated at 37°C or 40°C. Swimming motility was photographed at 7 h post inoculation. **(C)** THP-1 human monocyte differentiated into macrophage-like cells were infected with *S*. Paratyphi A WT and its isogenic Δ*fliC*, Δ*hilE* and Δ*hilE fliC* null mutant strains at MOI of one. Bacterial uptake was determined as in (A). Columns show the mean of eight independent infections, with SEM represented by the error bars. Unpaired, 2-tailed Student *t* test was used to determine statistical significance relative to *S*. Paratyphi A WT. *, P-value <0.05; **, P-value <0.01; ***, P-value < 0.001; ns, not statistically significant.

(39–42°C), motility, epithelial cell invasion, and uptake by macrophages are markedly attenuated in all typhoidal *Salmonella* serovars, but not in NTS [10]. However, the molecular mechanism responsible for this dichotomy remained unknown. Here, we show that the impaired flagella-mediated motility of *S*. Paratyphi A under elevated temperature concurs with

reduced transcription of multiple flagellar and chemotaxis genes from classes 1, 2, and 3, and lower expression of motility and chemotaxis proteins, as demonstrated by RT-qPCR, proteomics analyses, and motility assays on soft agar. Using an unbiased Tn mutagenesis approach in a *S*. Paratyphi A reporter strain, we found that HilE, a negative regulator of SPI-1 genes [35,36], plays a key role in the motility thermoregulation of *S*. Paratyphi A in a HilD-dependent manner. Since it has been previously shown that HilE regulates HilD activity through protein-protein interactions by preventing HilD homodimerization [16] and preventing its binding to the *flhDC* promoter [17], our working model suggests that motility thermoregulation in *S*. Paratyphi A is based on the transient interaction of HilE with HilD. We speculate that in typhoidal *Salmonella* under high physiological temperatures, HilD binding to the *flhDC* promoter is inhibited by HilE with the possible involvement of other factors (see below) and the expression of the entire motility regulon is repressed (Fig 8).

A previous study has already reported the role of HilE in *Salmonella* pathogenicity and physiology by showing that HilE is required to reduce the growth cost imposed by the expression of HilD-regulated genes, and that HilE is necessary for successful intestinal colonization by *S*. Typhimurium [37]. HilE expression was also shown to be positively controlled by the virulence and motility regulators SirA/BarA [37] and FimZ [38], respectively, as well as by the H-NS antagonist LeuO [39]; it was also shown to be negatively controlled by the global carbohydrate metabolism regulator, Mlc [40]. Such complex and tight regulation of HilE highlights its central role in the global regulatory network of *Salmonella*.

Interestingly, thermoregulation by HilE was found to be common in the three typhoidal serovars (Typhi, Paratyphi A, and Sendai), but not in *S*. Paratyphi B, which did not exhibit a thermoregulated motility and presented a phenotype that was more similar to the motility behavior of the NTS serovars. While *S*. Paratyphi B infection can result in a systemic disease, this serovar can also cause a NTS-typical gastroenteritis disease in humans [41,42]. Moreover, previous mouse infection studies with *S*. Paratyphi B have shown that, unlike other typhoidal serovars, this pathogen can establish a

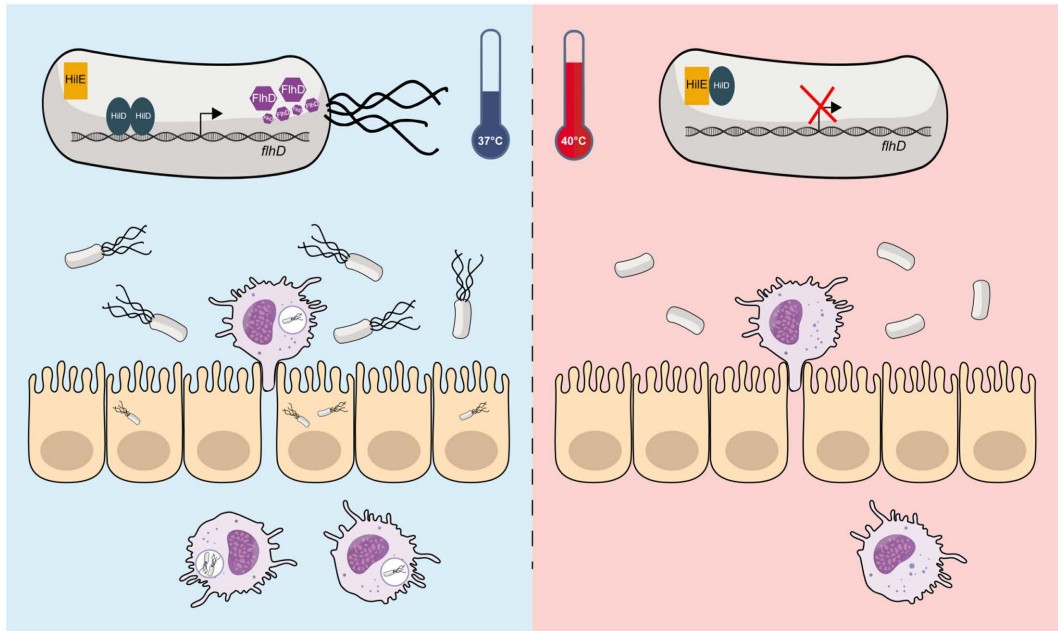

**Fig 8. Graphical model illustrating the role of HilE in Typhoidal *Salmonella* motility. (Left Panel)** at 37°C, the expression of FlhD is induced through the binding of HilD to *the flhDC* promoter. High expression of FlhDC leads to upregulation of the entire flagellar-motility regulon, *Salmonella* uptake by macrophages is facilitated and pathogen dissemination occurs. **(Right panel)** at elevated physiological temperature, the HilD-HilE interaction inhibits *flhDC* transcription and the flagellar-motility regulon is downregulated. Under these conditions, *Salmonella* uptake by macrophages is impaired, in a way that can restrict systemic dissemination of the pathogen.

lethal disease in mice following intraperitoneal infection, suggesting that *S.* Paratyphi B is less restricted to the human host than other enteric fever causing serovars [42]. Consistent with its classification as an atypical typhoidal serovar, our findings indicated that *S.* Paratyphi B (reference strains SARA 43 and SARA 44) regulates motility differently than the other typhoidal serovars Typhi, Paratyphi A, and Sendai. Furthermore, while HilE was shown to control thermoregulation in typhoidal *Salmonella*, our results demonstrated that HilE is not involved in thermoregulation in NTS serovars. This suggests that typhoidal and non-typhoidal *Salmonella* utilize distinct regulatory mechanisms to coordinate their motility under physiologically relevant elevated body temperature. This difference may reflect adaptation to discrete infection strategies and host interactions in typhoidal vs. NTS strains.

During infection, *Salmonella* invades host macrophages, which serve as its preferred replicative niche and act as "Trojan horses" to transport the bacteria into systemic tissues [18]. Previous studies from our lab found that functional flagella are essential for *Salmonella* entry into phagocytic cells, as flagella-deficient mutants showed significantly reduced macrophage entry despite unchanged adherence [10]. Here, we showed that under elevated physiological temperatures, when the motility of typhoidal *Salmonella* is reduced, macrophage entry by *S.* Paratyphi A is also impaired; furthermore, the deletion of *hilE*, not only increased *S.* Paratyphi A motility, but also enhanced its uptake by human macrophages at 40°C. Since motility thermoregulation by HilE was found to be common to serovars Paratyphi A, Typhi, and Sendai, but not to prevalent NTS strains, it is tempting to suggest that HilE plays a role in directing typhoidal *Salmonella* infection during the acute phase of the disease. At this stage of the infection, when fever symptoms are presented and the body temperature of enteric fever patients reaches 39–40°C, typhoidal *Salmonella* are typically already present in the blood stream and in systemic sites. Pathogen response to this cue and subsequent repression of its motility and flagellin expression can lower its own immunogenic signature and detection by Toll-like receptor 5 (TLR-5)-expressing cells [43]. However, additional benefits could be a reduced entry into phagocytic cells and limiting excessive spread to systemic sites, while supporting intestinal residing, which facilitate shedding and transmission.

It is interesting to note that a similar phenotype of temperature-dependent down-regulation of flagellar motility and a role in pathogenicity has also been described in *Listeria monocytogenes* [44] that shares with *Salmonella* a similar lifestyle as a facultative intracellular pathogen. Although, the underlying mechanism responsible for the motility thermoregulation in both bacteria is very different, this can provide an intriguing example of convergent evolution of pathogen adaptation to its human host.

The relationship between motility and virulence in *Salmonella* extends beyond mere locomotion. Several regulatory proteins, including FlhD, FlhC, and FliZ, function as dual regulators of both flagellar biosynthesis and virulence gene expression [45–47]. This regulatory crosstalk creates a coordinated network linking motility with virulence programs, particularly those governed by the SPIs. Our findings highlight prominent regulatory differences between typhoidal and NTS and reveal the key role of HilE in motility thermoregulation in typhoidal *Salmonella*. Moreover, these results show how HilE-mediated thermoregulation affects flagellin expression, flagella-mediated motility, and uptake by macrophages, in a way that may fine-tune the dissemination of typhoidal *Salmonella* to systemic sites by macrophages [48] and innate immune response facilitated by TLR-5 signaling [43], under febrile disease conditions.

Although we showed that HilE mediates motility thermoregulation in *S.* Paratyphi A, but not in *S.* Typhimurium, it is likely that additional factor than merely HilE is involved in this regulatory pathway. In other words, we believe that HilE is required, but not sufficient to coordinate motility thermoregulation in *S.* Paratyphi A. This hypothesis is based on the observation that HilD share the same sequence in *S.* Paratyphi A and *S.* Typhimurium (S5A Fig), HilE in *S.* Paratyphi A and *S.* Typhimurium although not identical is highly conserved (S5B Fig) and that swapping HilE from *S.* Paratyphi A and *S.* Typhimurium was not sufficient to change motility thermoregulation in either serovar (S5C Fig).

Interestingly, a deletion of a small 49-bp region at the P5 *flhDC* promoter, downstream to the HilD binding site, we named CITRE (standing for cis thermoregulatory element; S6A Fig) has also led to the loss of motility thermoregulation in *S.* Paratyphi A, but not in *S.* Typhimurium as did the absence of HilE, on the transcriptional level (S6B Fig), protein level

(S6C Fig) and phenotypic level (S6D Fig). Because CITRE was predicted to contain two binding sites of the nucleoid-associated proteins H-NS, known to act as a transcriptional repressor [49] and as a HilD antagonist [50], we speculate that additional regulatory factors (possibly H-NS), which binds to this loci, are also required in conjunction with HilE to orchestrate motility thermoregulation in *S*. Paratyphi A.

Future research in this direction is expected to shed light on the involvement of H-NS in motility regulation in *S*. Paratyphi A, and reveal how HilE-HilD interaction changes under different physiological temperatures. It is also of interest to further study how other genes identified in the transposon screen (*dapB*, *dapE*), involved in the lysine/ diaminopimelate (DAP) biosynthetic pathway, may contribute to motility thermoregulation of typhoidal *Salmonella*. Additional mechanistic insights into motility thermoregulation in typhoidal *Salmonella* will improve our understanding of the strategies employed by this pathogen in adaptation to the human host and illuminate how this behavior contributes to the development of enteric fever.

## Materials and methods

### Bacterial strains and culture conditions

Strains and plasmids used in this study are listed in S1 Table. Bacterial cultures were routinely maintained in Lennox Luria-Bertani (LB) broth. When appropriate, the medium was supplemented with antibiotics in the following concentrations: kanamycin, 50 µg/ml; ampicillin, 100 µg/ml; and nalidixic acid, 20 µg/ml. For the *lacZ* reporter screens 40 µg/ml of 5-Bromo-4-chloro-3-indolyl-β-D-galactopyranoside (X-gal) and 0.5 mM isopropyl-β-D-thiogalactopyranoside (IPTG) were spread onto LB-agar plates.

### Molecular biology and cloning

All primers used in this study are listed in S2 Table. Oligonucleotides were purchased from IDT and PCR was carried out using Phusion Hot Start Flex DNA Polymerase (New England BioLabs) or with Red Load Taq Master (Larova). None-polar Δ*hilE*, Δ*hilD*, Δ*dapB* and Δ*fliC* null mutants and a 49-bp deletion of CITRE sequence in various *S. enterica* backgrounds were constructed using the λ Red–mediated recombination system and the resistance cassette was then eliminated from the genome by using a helper plasmid encoding the FLP recombinase [51]. For complementation of Δ*hilE* and Δ*hilD* in *S*. Paratyphi A and *S*. Typhimurium, the intact sequence was PCR amplified using gDNA from *S*. Paratyphi A 45157 and *S*. Typhimurium SL1344 respectively as a template. The obtained amplicons of *hilE* and *hilD* resulted in 815 and 1245 bp amplicons, respectively that included their native regulatory regions. *hilE* was amplified using primers "*hilE* SacI F" and "*hilE* BamHI R". *hilD* was amplified using primers "*hilD* SacI F" and "*hilD* BamHI R", both fragments were digested with SacI and BamHI and cloned into the low copy-number vector, pWSK29. To construct the motility reporter systems the *flhB*, *fliC*, *motA* and *flhD* promoters were PCR amplified using *S*. Paratyphi A 45157 gDNA as a template, which resulted in 219, 379, 153, and 730-bp DNA fragments that included their native upstream regulatory regions and the first 7–9 amino acid codons of the target gene. The promoter of *flhB* was amplified using primers "*flhB*p SmaI F" and "*flhB*p BamHI R"; *fliC* with "*fliC*p SmaI F" and "*fliC*p BamHI R"; *motA* with the primers "*motA*p SmaI F" and "*motA*p BamHI R"; and *flhD* was amplified with the primers "*flhD*p SmaI F" and "*flhD*p BamHI R". All fragments were digested with SmaI and BamHI and cloned into pMC1403 upstream to a promoterless *lacZ*Y gene.

### Transposon insertion library

A Tn10 transposon mutagenesis library was constructed in a nalidixic acid resistant *S*. Paratyphi A 45157 strain, harboring the motility reporter system *flhB*::*lacZ* cloned in pMC1403. The transposon library was generated by conjugating the pJA1 plasmid harboring a mini-Tn10 transposon containing a kanamycin resistance marker and a very low insertional bias [52]. pJA1 plasmid was conjugated into the recipient *S*. Paratyphi A strain, using an *E. coli* SM10 λ pir as the donor strain. Transconjugant cells were selected and induced overnight in LB supplemented with 0.8 M IPTG and plated on LB agar

plates supplemented with X-gal and ampicillin (to select for reporter plasmid), kanamycin (to select for the transposon) and nalidixic acid (to select for the recipient *S.* Paratyphi A strain). The plates were then incubated at 40°C and blue colonies that indicated induced expression of *flhB::lacZ* were selected for further phenotypic characterization.

### Identification of the transposon insertion sites

Ttransposon insertion sites were determined by two stages of PCR amplification according to [52] with the following modifications. DNA flanking the insertion sites was amplified using GoTaq DNA Polymerase (Promega) in a reaction buffer containing 25 mM $MgCl_2$, using a primer that binds the middle part of the kanamycin resistance gene in pJA1 (primer nptII_A) and an 'arbitrary' primer that randomly binds in the genome (primer 1366-Arb-1B). The PCR protocol included an initial denaturation step at 95°C for 3 min followed by 35 cycles of: 92°C for 15 s, 38°C for 30 s, and 72°C for 2 min, and a final elongation step of 72°C for 5 min. PCR products were treated with exonuclease I (New England Biolabs), and shrimp alkaline phosphatase (New England Biolabs) at 37°C for 15 min then inactivated at 85°C for 15 min. Then, a second PCR step was applied using the transposon-specific primer nptII_B that binds just downstream of the nptIIA primer and the 1368-Arb-1 primer, which binds to primer 1366-Arb-1B, used in step 1. A second PCR amplification included an initial denaturation step at 95°C for 3 min followed by 35 cycles of 92°C for 15 s, 58°C for 30 s, and 72°C for 2 min, and a final elongation step at 72°C for 5 min. The obtained amplicons ranged between 200 and 1500 bp were purified from 1% agarose gel, using the NucleoSpin Gel and PCR Clean-up kit (Macherey-Nagel) and subjected to a Sanger sequencing, using the nptII_C primer.

### β-galactosidase assays

*S.* Paratyphi A and *S.* Typhimurium strains expressing the *flhB::lacZ* or the *flhD::lacZ* reporter constructs were grown in LB supplemented with ampicillin overnight at 37°C, subcultured (1:100) and grown under aerobic conditions for 3 h at 37°C or 40°C. To determine the β galactosidase specific activity 500 µl of cultures were added to 500 µl Z-Buffer (0.06 M $Na_2H$-$PO_4 \cdot 7H_2O$, 0.04 M $NaH_2PO_4 \cdot H_2O$, 0.01 M KCl, 0.001 M $MgSO_4 \cdot 7H_2O$, and 0.05 M β-mercaptoethanol; pH 7) and assayed as detailed in [53], using o-nitrophenyl-β-D-galactopyranoside (ONPG, Sigma-Aldrich) as a substrate.

### Quantitative reverse transcription polymerase chain reaction (RT-qPCR)

*S.* Typhimurium and *S.* Paratyphi A cultures were grown overnight at 37°C, subcultured (1:100 dilution) and grown under aerobic conditions for 3 h at 37°C or 40°C. RNA was extracted from 400 µl of cultures using the RNA protect Bacteria Reagent (QIAGEN) and RNeasy mini kit (QIAGEN) according to the manufacturer's instructions. To diminish any genomic DNA contamination, RNA was treated with TURBO DNA-free Kit (Thermo Fisher Scientific). The quantity and quality of the extracted RNA were determined by Nanodrop 2000c (Thermo Fisher Scientific). cDNA was synthesized using qScript cDNA Synthesis Kit (Quanta-bio) in a T100 thermal cycler by (Bio-Rad Laboratories). Real-Time PCR reactions were performed in a StepOne Real-Time PCR (Applied Biosystems) instrument. Each reaction was carried out in a total volume of 20 µl in a 96-well optical reaction plate (Applied Biosystems). Melting curve analysis verified that each reaction amplified a single PCR product. Data were normalized to the housekeeping gene *16S* rRNA and analyzed by the $2^{-\Delta\Delta CT}$ method [54].

### Motility assay

Ten µL of overnight *Salmonella* cultures grown in LB broth at 37°C were placed onto 0.3% agar LB plates, the plates were incubated for 4.5–12 h at 37°C or 40°C without being inverted. The motility radius was measured with a ruler and imaged using a Fusion Solo X system (Vilber).

### Western blotting

Overnight cultures grown in LB at 37°C were diluted 1:100 and subcultured for 3 h at 37°C or 40°C under aerobic conditions. The cultures were $OD_{600}$ normalized, centrifuged, and the pellets were resuspended in 100 µL of 1 × sodium dodecyl

sulfate-polyacrylamide gel electrophoresis (SDS-PAGE) sample buffer (0.15 M Tris-HCl pH 6.8, 10% 2-Mercaptoethanol, 1.2% SDS, 30% Glycerol, 0.04% Bromophenol blue). Boiled samples were separated on 12% SDS-PAGE and transferred to a polyvinylidene fluoride (PVDF) membrane (Bio-Rad Laboratories) as we previously described [55]. Blots were probed with custom (Cusabio) polyclonal anti- *S.* Paratyphi A FliC antibody (diluted 1: 30,000) that were raised in rabbits against polypeptide synthetized based on the *S.* Paratyphi A FliC sequence (XBC74829.1). Goat anti-rabbit antibody conjugated to horseradish peroxidase (Abcam; ab6721, diluted 1: 3,000) was used as a secondary antibody, followed by detection with enhanced chemiluminescence (ECL) reagents (Bio-Rad Laboratories). For a loading control, membrane was sub-mersed for 10-30 seconds in 100% methanol, briefly rinsed in $dH_2O$ and stained with Ponceau S solution (Sigma-Aldrich). Stained membranes were then destained with $dH_2O$ and imaged using the Fusion solo X system (Vilber). Bands densitom-etry was determined by Image J as explained in [56].

## Mass spectrometry

*S.* Paratyphi A and *S.* Typhimurium cultures were grown in LB for overnight at 37°C and then subcultured 1:100 and grown for 3 h at 37°C or 40°C. Bacterial culture aliquots (1 ml) were centrifuged at 10,000 *g*, washed twice with phosphate buffered saline (PBS) and resuspended in 5% SDS, 100 mM TRIS and 10 mM DL-dithiothreitol (DTT). Mass spectrometry analysis was carried out at the Smoler Proteomic Center at the Technion, Haifa, Israel. Protein samples were re-boiled at 95°C for 5 minutes, sonicated, and centrifuged at 10,000 *g* for 10 minutes. Precipitation was performed using cold 80% acetone, and protein pellets were dissolved in 8.5 M Urea, 400 mM ammonium bicarbonate, and 10 mM DTT. Proteins were reduced, alkylated with iodoacetamide, and digested with trypsin (Promega) overnight at 37°C, followed by a sec-ond digestion for 4 h. Peptides were desalted using C18 StageTips, dried, and resuspended in 0.1% formic acid with 2% acetonitrile. The resulted peptides were analyzed by liquid chromatography with tandem mass spectrometry (LC-MS-MS) using a Q Exactive HF mass spectrometer (Thermo Fisher Scientific) fitted with a capillary HPLC (Thermo Fisher Sci-entific). The peptides were loaded in solvent A (0.1% formic acid in water) on a homemade capillary column (30 cm, 75-micron ID) packed with Reprosil C18-Aqua (Dr. Maisch GmbH, Germany). The peptides mixture was resolved with a 6–30% linear gradient of solvent B (80% acetonitrile with 0.1% formic acid) for 180 minutes followed by gradient of 15 minutes of 28–95% and 15 minutes at 95% solvent B at flow rates of 0.15 µl/min. Mass spectrometry was performed in a positive mode using repetitively full MS scan followed by high collision dissociation high collision dissociation (HCD) and dynamic exclusion.

The mass spectrometry data was analyzed using Proteome Discoverer 2.4 (Thermo) using Sequest search engine, searching against the *S.* Typhimurium (strain SL1344) and *S.* Paratyphi A (strain 45157) proteomes from the Uniprot database with mass tolerance of 20 ppm for the precursor masses and 0.02 Da for the fragment ions. Oxidation on methi-onine, and protein N-terminus acetylation were accepted as variable modifications and carbamidomethyl on cysteine was accepted as static modifications. Minimal peptide length was set to six amino acids and a maximum of two miscleavages was allowed. The data was quantified by label free analysis using the same software. Peptide- level false discovery rates (FDRs) were filtered to 1% using the target-decoy strategy.

## Human macrophages uptake assay

Human macrophages (THP-1) were obtained from ATCC and seeded at $2.5 \times 10^5$ and $1.2 \times 10^6$ cells/well in a 24-well and 6-well tissue culture dish, respectively. To differentiate THP-1 monocytes into macrophage-like cells, THP-1 cells were incubated at 37°C in a humidified atmosphere with 5% $CO_2$ in RPMI 1640 medium supplemented with 10 mM HEPES, 2 mM L-Glutamine, 0.1 mM MEM non-essential amino acids solution, 1 mM sodium pyruvate, 0.05 mM β-mercaptoethanol, 20% heat-inactivated fetal bovine serum (FBS), and 50 ng/ml phorbol 12-myristate 13-acetate (PMA) (Sigma-Aldrich). After 24 h, cells were washed and incubated in the same medium without PMA for additional 48 h. THP-1 infection exper-iments were carried using the gentamicin protection assay, as described elsewhere [57]. Briefly, differentiated THP-1

were infected with *S*. Typhimurium, *S*. Paratyphi A, *S*. Typhi and *S*. Sendai cultures that were grown in LB broth overnight at 37°C or 40°C under aerobic conditions, at multiplicity of infection (MOI) of 1–10. A mild centrifugation force (500 *g* for 5 min at 25°C) was applied immediately after bacteria were added the cells to synchronize the infection. Infected cells were subsequently incubated at 37°C or 40°C, and at 30 min post infection the cells were washed with PBS with $Ca^{2+}$ and $Mg^{2+}$ and further incubated in medium supplemented with 100 µg/ml gentamicin. At 2 h p.i. cells were washed and lysed with lysis buffer (1% Triton X- 100, 0.1% SDS in PBS with $Ca^{2+}$ and $Mg^{2+}$). *S*. Typhimurium, *S*. Paratyphi A, *S*. Typhi and *S*. Sendai uptake was determined by calculating the number of intracellular colony forming units (CFUs) at 2 h p.i. divided by the number of infecting bacteria.

## Supporting information

**S1 Fig. The flagella-chemotaxis regulation in *S*. Paratyphi A.** (**A**) The motility-chemotaxis regulon in *Salmonella* is organized in three hierarchical transcriptional classes. The class 1 operon contains the *flhD* and *flhC* genes that together encode the master regulator of flagella-chemotaxis regulon, FlhDC. The heteromultimeric complex ($FlhD_4C_2$) is positioned at the top of this hierarchy and activates the transcription of class 2 operons. Class 2 genes responsible for basal body assembly. Following the assembly of the early flagellar basal body, the repressor FlgM is exported from the cell, releasing the class 3 transcription factor $\sigma^{28}$ (FliA). This activates the expression of class 3 genes, leading to the completion of the hook and filament assembly. The positive regulators HilD, FliZ, FlhDC and FliA are shown in green, while the negative regulators involved FliT and FlgM are shown in red. Motility genes that were analyzed by RT-qPCR in Fig. 1A are shown in magenta. Genes that are inactivated in *S*. Paratyphi A are shown in light grey. (**B**) Graphical illustration of the flagella structure. Proteins that were detected by the MS analysis in Fig. 1B are shown in purple.
(PDF)

**S2 Fig. Reconstruction null deletion in *hilE*, *invA*, *invG*, and *dapB* and their effect on *S*. Paratyphi A motility.** (**A**) *S*. Paratyphi A WT and its isogenic null mutant strains Δ*hilE*, Δ*invG*, Δ*invA* and Δ*dapD* were grown overnight at 37°C, spotted onto soft agar plates and incubated at 37°C and 40°C for 5.5 h. The fold change motility at 40°C vs. 37°C is shown, while SEM is represented by the error bars. One-way ANOVA was used to determine statistical significance in relation to *S*. Paratyphi A WT. (**B**) Swimming motility on soft agar plates of *S*. Paratyphi A WT, its isogenic Δ*hilE* strain and two transposon insertion mutants in *hilE* (clones 3–2 and 8–2). All strains were grown overnight at 37°C and then 10 µl from each culture were spotted on soft agar LB plates that were incubated at 37°C and 40°C for 5.5 h. (**C**) The motility radius of the strains was measured and their fold change in motility at 40°C vs. 37°C is shown. One-way ANOVA was used to determine statistical significance relative to *S*. Paratyphi A WT. *, P-value <0.05; **, P-value <0.01; ***, P-value < 0.001; ns, not statistically significant.
(TIF)

**S3 Fig. Motility thermoregulation is not common among NTS serovars.** The motility of three typhoidal (Typhi, Paratyphi A, and Sendai) and 16 NTS serovars was measured on semisolid LB agar plates at 37°C and 42°C. The mean change in the motility between 42°C and 37°C is shown. Bars represent the mean of 2–3 independent experiments and SEM is indicated by the error bars.
(PDF)

**S4 Fig. *S*. Paratyphi A uptake by THP-1 macrophages with and without application of a mild centrifugation.** Cultures of *S*. Paratyphi A (WT and its isogenic *hilE* and *fliC* mutants) were grown at 37 or 40°C and were used to infect differentiated THP-1 human macrophage-like cells at MOI of five at both temperatures. *Salmonella* uptake by THP-1 cells was determined using the gentamicin protection assay and calculated the percentage of the intracellular CFUs recovered at 2 h p.i from the total number of CFUs used to infect the cells. Bars present the mean value four biological repeats and thier calculated SEM. Unpaired, 2-tailed Student *t* test was used to calculate statistical significance.
(PDF)

**S5 Fig. Swapping HilE between *S*. Paratyphi A and *S*. Typhimurium does not alter motility thermoregulation.** (**A**) The 310 amino acid protein sequence of HilD in *S*. Typhimurium strain SL1344 (accession number CBW18953.1) and its ortholog in *S*. Paratyphi A strain 45157 (accession number QWV88109.1) were aligned using CLUSTALW. Alignment image of the amino acid identity was created by the BoxShade tool (https://junli.netlify.app/apps/boxshade/) and show complete identity at the protein level between these two orthologues. (**B**) 177 amino acids of the HilE protein in *S*. Typhimurium strain SL1344 (accession number CBW20528.1) and its ortholog in *S*. Paratyphi A strain 45157 (accession number QWV89528.1) were aligned and presented as above. Three divergent amino acids between these serovars are highlighted. (**C**) *S*. Typhimurium strains including WT, Δ*hilE*, Δ*hilE* complemented with the empty vector (Δ*hilE*/P), *hilE* from *S*. Typhimurium (Δ*hilE*/*hilE*$_{STM}$) or *hilE* from *S*. Paratyphi A (Δ*hilE*/*hilE*$_{SPA}$); and *S*. Paratyphi A strains including WT, Δ*hilE*, and Δ*hilE* complemented with the empty vector (Δ*hilE*/P), *hilE* from *S*. Paratyphi A (Δ*hilE*/*hilE*$_{SPA}$) or *S*. Typhimurium (Δ*hilE*/*hilE*$_{STM}$) were grown at 37°C aerobically for overnight. Ten μl from each culture were spotted on soft LB agar plates that were incubated at 37°C or 40°C for 4 h before imaging.
(PDF)

**S6 Fig. Identification of a CIS thermoregulatory element (CITRE) contributes to motility thermoregulation in *S*. Paratyphi A.** (**A**) The genetic organization of the *flhDC* promotor is shown. HilD binding site is presented according to Singer *et al*. [17]. Black arrows indicate the six *flhDC* transcription start sites, while their -10 promoter elements are indicated by a closed box, as reported by Yanagihara *et al*. [58]. The HilD binding site is highlighted in blue and the identified 49-bp CITRE element is marked by a green bar, overlapping with two putative H-NS binding sites predicted by PRODORIC (https://www.prodoric.de) that are shown as orange boxes. (**B**) WT *S*. Typhimurium, WT *S*. Paratyphi A, and their isogenic strains harboring a 49-bp deletion of the CITRE element (ΔCITRE) were subcultured in LB grown for 3 h at 37°C and 40°C and subjected to RT-qPCR analysis. Fold change in the expression of five flagellar genes *(flhD, fliL, flgK, motA,* and *fliC)* at 40°C relative to their expression at 37°C is shown as the mean value of three biological replicates, with SEM represented by the error bars. Unpaired, 2-tailed Student *t* test was used to determine statistical significance between the WT background and its isogenic ΔCITRE mutant. (**C**) The level of FliC at the cellular fraction was analyzed by Western blotting using a *S*. Paratyphi A anti-FliC antibody. Ponceau S staining and the detection of a ~34 kDa cross-reacting band were used as loading controls. FliC expression in the ΔCITRE strain, relative to its expression in the WT background at 37°C, is presented by numerical values below the WB. (**D**) Cultures of the above strains were spotted onto soft agar plates that were incubated at 37 and 40°C, for 5.5 h. The fold change in motility at 40°C relative to 37°C is shown, while SEM is represented by the error bars. Unpaired, 2-tailed Student *t* test was used to determine statistical significance. **, P-value <0.01; ***, P-value <0.001; ns, not statistically significant.
(TIF)

**S1 Table. Bacterial strains and plasmids used in this study.**
(DOCX)

**S2 Table. Primers used in this study.**
(DOCX)

**S3 Table. The numerical values used to build graphs.**
(XLSX)

## Acknowledgments

We thank Dr. George Church from the Department of Genetics, Harvard Medical School for the kind gift of the EPEC SM10λpir strain harboring the pJA1 transposon plasmid. We are thankful to Dr. Nikki Freed from University of Auckland for helpful communication and practical suggestions regarding the identification of transposon insertion sites. We thank

Dr. Dana Elhadad for laying the foundation to this study and her contribution to the motility assays. We are also grateful to Shani Shem-Tov for her valuable help with the artwork, and for Dr. Dor Salomon for critical reading on the manuscrpt.

## Author contributions

**Conceptualization:** Ohad Gal-Mor.

**Data curation:** Rivka Shem-Tov, Ohad Gal-Mor.

**Formal analysis:** Rivka Shem-Tov, Ohad Gal-Mor.

**Funding acquisition:** Ohad Gal-Mor.

**Investigation:** Rivka Shem-Tov, Ohad Gal-Mor.

**Resources:** Ohad Gal-Mor.

**Supervision:** Ohad Gal-Mor.

**Validation:** Ohad Gal-Mor.

**Visualization:** Rivka Shem-Tov, Ohad Gal-Mor.

**Writing – original draft:** Rivka Shem-Tov, Ohad Gal-Mor.

**Writing – review & editing:** Ohad Gal-Mor.

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
