## [Decision Letter · Decision Letter 0]

4 Jun 2025

PPATHOGENS-D-25-00929

HilE mediates motility thermoregulation in typhoidal Salmonella serovars at elevated physiological temperatures

PLOS Pathogens

Dear Dr. Gal-Mor,

Thank you for submitting your manuscript to PLOS Pathogens. After careful consideration, we feel that it has merit but does not fully meet PLOS Pathogens's publication criteria as it currently stands. Therefore, we invite you to submit a revised version of the manuscript that addresses the points raised during the review process.

Please submit your revised manuscript within 30 days Aug 03 2025 11:59PM. If you will need more time than this to complete your revisions, please reply to this message or contact the journal office at plospathogens@plos.org. Please include the following items when submitting your revised manuscript:

We look forward to receiving your revised manuscript.

Kind regards,

Andreas J Baumler

Academic Editor

PLOS Pathogens

Matthew Wolfgang

Section Editor

PLOS Pathogens

Sumita Bhaduri-McIntosh

Editor-in-Chief

PLOS Pathogens

orcid.org/0000-0003-2946-9497

Michael Malim

Editor-in-Chief

PLOS Pathogens

orcid.org/0000-0002-7699-2064

**Journal Requirements:**

At this stage, the following Authors/Authors require contributions: Rivka Shem-Tov, and Ohad Gal-Mor. Please ensure that the full contributions of each author are acknowledged in the "Add/Edit/Remove Authors" section of our submission form.

2) We have noticed that you have uploaded Supporting Information files, but you have not included a list of legends. Please add a full list of legends for your Supporting Information files after the references list.

3) Some material included in your submission may be copyrighted. According to PLOSu2019s copyright policy, authors who use figures or other material (e.g., graphics, clipart, maps) from another author or copyright holder must demonstrate or obtain permission to publish this material under the Creative Commons Attribution 4.0 International (CC BY 4.0) License used by PLOS journals. Please closely review the details of PLOSu2019s copyright requirements here: PLOS Licenses and Copyright. If you need to request permissions from a copyright holder, you may use PLOS's Copyright Content Permission form.

Potential Copyright Issues:

- Please confirm (a) that you are the photographer of Figures 1C, 2A, 2B, 3A, 3B, 4C, 7B, and S2B., or (b) provide written permission from the photographer to publish the photo(s) under our CC BY 4.0 license.

4) Please ensure that the funders and grant numbers match between the Financial Disclosure field and the Funding Information tab in your submission form. Note that the funders must be provided in the same order in both places as well.

**Reviewers' Comments:**

Reviewer's Responses to Questions

**Part I - Summary**

Reviewer #1: This manuscript describes in an over complicated fashion the mechanistic explanation for a previously recognised thermoregulation of flagellar gene expression in Salmonella serovars. Embedded deep in this dialogue is a further interesting observation with respect to the impact the thermoregulation has upon macrophage uptake. This second observation is in some essence more important than defining the role HilE plays in controlling flagellar gene expression. It is the combination of these two points that I can understand why the authors have submitted this paper. However, from my point of view as a reviewer there is substantial work to be done on the data presentation to allow a reader to easily follow their rationale for the experiments performed and the analysis of their data.

I am, however, confident that the authors can recognise the constructive approach of this review and in doing so generate a nice narrative of two/three observations for Salmonella as a species. They show that HilE controls thermoregulation of S. Paratyphi, not all serovars exhibit this phenotype and that it has a marked impact on macrophage uptake. Please consider restructuring the results based on this order.

Reviewer #2: Shem-Tov and Gal-Mor identified in their manuscript the SPI-1 repressor HilE as a negative regulator of motility at elevated temperatures for typhoidal Salmonella serovars. While the group described the inhibition of bacterial motility at 42°C for specific for typhoidal serovars in an earlier manuscript, they reproduced their finding in the present manuscript at 40°C more closely resembling enteric fever in patients. To elucidate the mechanism behind, the group designed a chromogenic reporter system allowing for high throughput screening of flagellar gene induction. This reporter system was combined with a dense transposon screen to identify genes responsible for the attenuated motility at febrile temperatures in a multi-stage screening setup. Here, genes involved in lysine production (yffB, dapBE) and SPI-1 related genes (invG, sipD, hilE) were identified. Functional characterization of knockout mutants identified HilE to be responsible for the observed phenotype. Since HilE is described as a negative regulator of the SPI-1 master regulator HilD, the group also tested hilD deletion mutants in reporter, motiliy and qPCR assays which demonstrated the HilE effect to be mediated via HilD. HilD probably acts via transcriptional activation of flagellar master regulator FlhDC. Moreover, the group tested the uptake of S. Paratyphi A (SPA) in human THP-1 macrophages. Absence of HilE led to increased uptake of SPA at 40°C. Finally they tested different other typhoidal and non-typhoidal serovars (NTS) for the motility phenotype at 40°C. Interestingly, only typhoidal Salmonella with the exception of S. Paratyphi B showed the reduced motility at 40°C. Deletion of hilE in the typhoidal serovars SPA, Typhi and Sendai increased motility at 40°C.

The manuscript is clearly and well written making the argumentation of the authors comprehensible. However, although identifying the HilE-HilD axis to be involved in thermoregulation of motility, an actual temperature-dependent mechanism remains elusive. In their working model, HilE should have an increased affinity for HilD in typhoidal Salmonella (except SPB) compared to NTS (and SPB). Can this be explained on the molecular level and ideally addressed experimentally? Although I can follow the argumentation of the authors to include the infection data of THP-1 macrophages (Fig. 6, systemic spread in enteric fever), it still feels a little disconnected from the rest of the manuscript.

**Part II – Major Issues: Key Experiments Required for Acceptance**

Reviewer #1: No further experiments are necessary. However, The data in figure 6 would benefit similar mutants in other Salmonella serovars being tested and included in this analysis. Possibly something to think of for the future maybe.

Reviewer #2: 1. Can the authors demonstrate the temperature-dependent difference of the HilD-HilE interaction using co-IP or suitable reporter systems (e.g. as shown in Grenz et al. J. Bacteriol 2018 https://doi.org/10.1128/jb.00750-17)

2. Are there differences in the sequences of HilD and/or HilE which might explain this different behavior? If differences exist, do typhoidal Salmonella separate from NTS (and SPB) in unsupervised clustering of sequence alignments? Can critical amino acids be identified based on the known structure of the HilD-HilE complex?

3. Could there be temperature- and serovar dependent differences in HilE/D protein stability?

4. How does STM behave in the motility assay expressing HilE and/or HilD of STA and vice versa?

5. Based on their findings, there should be a major impact of SPI-1 T3SS expression at 40°C vs. 37°C in typhoidal Salmonella. Did the authors also test invasion of non-phagocytic cells under these conditions?

6. Although the group previously showed that entry in macrophages depends on functional flagella, it would be interesting to dissect the influence of motility/chemotaxis (near surface swimming) from flagella-mediated adhesion. Here, the authors could compare infection with (as in the present manuscript) and without mild centrifugation.

7. I’m wondering about the expression of hilE and hilD in O/N cultures. Depending on the conditions (aeration, NaCl content) SPI-1 expression could be rather low. Did the authors check with RT-qPCR expression of these genes (and flagellar genes)?

**Part III – Minor Issues: Editorial and Data Presentation Modifications**

Reviewer #1: Outlined below are my specific comments.

1. Please refrain from using STM, SPA etc it is not difficult to write S. Typhimuirum and S. Paratyphi A in its place and really helps the reader follow which serovar you are using.

2. Please rename all mutants as ∆hilE, ∆hilD etc etc instead of hilE and hilD, this will then conform with standard genetic practice and allow the reader to differentiate between you discussing a gene, its overexpression, its expression and its deletion.

3. Please place the supplementary figures into the supplementary document to go along with their legends. Having a figure with no legend in such material is pointless.

4. Figure Y-axis: I had a hard time understanding your y-axis choice for many of the graphs and thus the interpretation of most of the data. It would be appreciated if you follow the rules of using a uniform axis ( -2 to +2) and not have the X-axis cross the Y-axis at 1 when in essence these are normal bar charts. For example, the data in figure 1 suggests a very strong downregulation of motility when in reality S. Paratyphi A is MOTILE at 40°C but I agree is reduced compared to S. Typhimurium. As a result, the data in A and B does not really correlate with the phenotype observed which is a 3-fold reduction in motility as assayed by the semi-solid agar swarming assay. This will be a recurring theme that I am concerned that your expression data at times does not correlate well with the observed phenotype.

5. Figure 3: You do not need to convince a microbiologist/microbial geneticist that the colonies you have isolated were blue as shown in Fig 3A&B. If you really want to include this data place it in the supplementary data and edit out the text on page 10 lines 195 to 205.

6. Page 10 Genetics: Please justify why the numbers of mutants that are analysed keeps changing or present the data for all 23. You can have a clear statement that for this study the focus was HilE at the end of this genetic section, but please stay consistent. Please also make sure you numbering is consistent in Figure 3 you have these as 3-3 3-2 13-3 etc etc but in table 1 they are 3-15 3-19 etc etc which way around are they supposed to be? You must have sequence data for all of them so why are you only showing these.

7. Figure 3D, 4D, 5D, 7A and 7C. You are not presenting ‘relative motility data’ in these graphs. Furthermore, these graphs specifically relate back to point 3 regarding the irregular use of y-axis scales, the overexaggerated axis scale does not represent the true impact of these genetic manipulations.

I could not generate from the methods or my own assessment of your motility data shown in 3C how you came to these numbers. It is not simply the radius or diameter of the swarms divided by each other, and please do not use the argument that the 3D image is just an example, you are showing data that should in some context correlate to the quantification. I tried to do this using a ruler from your figures. For example, WT 37°C 41 mm 40°C 28 mm which equates to 0.68 or a 1.5 fold reduction and not ~0.3. Your data is robust and in every case is justified, just please consider presenting the reduction in an easier understandable format. Even if it is the quantified means placed next to each other on a bar chart without any convoluted further calculations.

8. Figure 4: The Ponceau S staining is not necessary in any of the figures or again if you feel it is put it in supplementary data. Furthermore, once you have defined your motility quantification method and presentation style you do not need to always show the motility swarms. In this instance as well as other places the motility swarm data is not correlating well with expression or quantification. Figure 4A please state what scale format is being used on the y-axis or again just use a linear scale so the reader can see how small some of the values are.

9. Page 15 297-318: You could discuss this data in a much simpler manner. You also overlook either purposefully or non-intentionally the impact of ∆hilE at 37°C.

10. Figure 6 please reassess your fold reduction statements or at least make it clearer what is being compared. For example, ∆hilE @ 40°C is ~ 150% ∆hilE @ 37°C is ~75% that’s a 2-fold difference not a 3-fold difference (page 17 line 358).

11. Figure legends: There is too much methodology hidden within the legends. If this is not in your methods section add it to the methods. If it is in the methods do not repeat it here. Please also explain what is meant in the methods by four biological repeats in one representative, out of two independent experiments is this n= 4 or n = 8?

12. Figure 7 Legend: Why were different strains measured at different times and why is this important enough for you to mention it? As you have mentioned it you really need to justify why in the main text or in the legend.

13. Page 23 line 478 This is the first EVER mention of SARA 43 and SARA 44. If there is a reason you used both, this needs justifying and possible highlighting in the results section when narrating through figure 7.

14. Discussion: I liked the discussion and how it correlates to how Salmonella as species interacts with us. I did not look at figure 8 although I did note it was nicely presented.

Reviewer #2: Minor point: Instead of using bars only for the plots, it is generally recommended to show in addition the individual data points making up that bar.

PLOS authors have the option to publish the peer review history of their article (what does this mean? ). If published, this will include your full peer review and any attached files.

**Do you want your identity to be public for this peer review?** For information about this choice, including consent withdrawal, please see our Privacy Policy .

Reviewer #1: No

Reviewer #2: No

**Figure resubmission:**
---

## [Decision Letter · Decision Letter 1]

16 Sep 2025

PPATHOGENS-D-25-00929R1

HilE mediates motility thermoregulation in typhoidal Salmonella serovars at elevated physiological temperatures

PLOS Pathogens

Dear Dr. Gal-Mor,

Thank you for submitting your manuscript to PLOS Pathogens. After careful consideration, we feel that it has merit but does not fully meet PLOS Pathogens's publication criteria as it currently stands. Therefore, we invite you to submit a revised version of the manuscript that addresses the points raised during the review process.

Please submit your revised manuscript within 30 days Nov 15 2025 11:59PM. If you will need more time than this to complete your revisions, please reply to this message or contact the journal office at plospathogens@plos.org. Please include the following items when submitting your revised manuscript:

We look forward to receiving your revised manuscript.

Kind regards,

Matthew C. Wolfgang, Ph.D.

Section Editor

PLOS Pathogens

Matthew Wolfgang

Section Editor

PLOS Pathogens

Sumita Bhaduri-McIntosh

Editor-in-Chief

PLOS Pathogens

orcid.org/0000-0003-2946-9497

Michael Malim

Editor-in-Chief

PLOS Pathogens

orcid.org/0000-0002-7699-2064

**Additional Editor Comments:**

Thank you for your thorough revision. Both reviewers were very positive, however, Reviewer 1 still has some outstanding concerns about the presentation of Figures 1B, 3C/D, 4D, 5D and 6A/C. Please specifically address this concern in your next and hopefully final revision. No additional comments need to be address.  

**Journal Requirements:**

**Reviewers' Comments:**

Reviewer's Responses to Questions

**Part I - Summary**

Reviewer #1: (No Response)

Reviewer #2: With the revised manuscript the authors have responded thoroughly and constructively to the previously raised critical points.

**Part II – Major Issues: Key Experiments Required for Acceptance**

Reviewer #1: The authors have made a good effort to cover all the comments from both reviewers. This is a complex story with essentially too much information/data integrated into one manuscript but I do get why they have done this.

The hilE swap and express experiments are a nice inclusion. However, they do seem to have opened up another research direction complicating the narrative even further.

Reviewer #2: My initial concerns focused on the mechanistic basis of the HilE/HilD effect, possible sequence differences, protein stability, and reciprocal expression of HilE/HilD between S. Typhimurium and S. Paratyphi A. These points have now been addressed. The authors show that HilD is identical between the two backgrounds, and HilE differs by only three amino acids. Protein swap experiments demonstrate that HilE is required but not sufficient for motility thermoregulation. They identified a novel cis-regulatory element in the flhD promoter, likely involving H-NS, which provides a plausible extension of the mechanism. The suggestion to differentiate motility/chemotaxis from flagella-mediated adhesion in macrophage uptake was implemented (new Fig. S4). With this new data the study is clearly strengthened including a refined working model.

**Part III – Minor Issues: Editorial and Data Presentation Modifications**

Reviewer #1: The quality of the new figures in the PDF we get to download as a reviewer is shocking, however, this is not entirely your fault. Many of the new data sets were over-pixlated so it was very difficult to read the axis.

You were asked to refrain from using the x-axis crossing at 1 in figures 1B, 3C/D, 4D, 5D and 6A/C. They are essenitally bar charts where your format over represents the negative phenotypes being observed and under represents any positive impact. You were also asked in the supplementary information to have the firgure legends with the figures instead of in seperate documents. I am unsure if this last point is PLOS pathogen policy but it makes it very difficult to appreciate the supplementary data flicking between documents. Would it not be easier to include all your supplementary data in one word/pdf file with each figure interleafed by its associated legend.

Reviewer #2: Uptake data are now presented in absolute rather than relative values, improving clarity. Sequence comparisons, promoter analyses, and additional serovar testing were also included. The reorganization of the results improved readability. Importantly, individual data points are now displayed in bar charts, which increases transparency.

However, some questions remain open including the precise temperature-dependent molecular mechanism. These results point to the likely involvement of additional regulatory factors and provide a clear perspective for future investigations.

PLOS authors have the option to publish the peer review history of their article (what does this mean? ). If published, this will include your full peer review and any attached files.

**Do you want your identity to be public for this peer review?** For information about this choice, including consent withdrawal, please see our Privacy Policy .

Reviewer #1: No

Reviewer #2: No

**Figure resubmission:**
---

## [Editor Report · Decision Letter 2]

29 Sep 2025

Dear Prof. Gal-Mor,

We are pleased to inform you that your manuscript 'HilE mediates motility thermoregulation in typhoidal Salmonella serovars at elevated physiological temperatures' has been provisionally accepted for publication in PLOS Pathogens.

Best regards,

Matthew C. Wolfgang, Ph.D.

Section Editor

PLOS Pathogens

Matthew Wolfgang

Section Editor

PLOS Pathogens

Sumita Bhaduri-McIntosh

Editor-in-Chief

PLOS Pathogens

orcid.org/0000-0003-2946-9497

Michael Malim

Editor-in-Chief

PLOS Pathogens

orcid.org/0000-0002-7699-2064
---

## [Editor Report · Acceptance letter]

Dear Prof. Gal-Mor,

We are delighted to inform you that your manuscript, "HilE mediates motility thermoregulation in typhoidal Salmonella serovars at elevated physiological temperatures," has been formally accepted for publication in PLOS Pathogens.

Best regards,

Sumita Bhaduri-McIntosh

Editor-in-Chief

PLOS Pathogens

orcid.org/0000-0003-2946-9497

Michael Malim

Editor-in-Chief

PLOS Pathogens

orcid.org/0000-0002-7699-2064